# SWAP-NAS: Sample-Wise Activation Patterns for Ultra-fast NAS

**Yameng Peng**[†]  **Andy Song**[†*]  **Haytham M. Fayek**[†]  **Vic Ciesielski**[†]  **Xiaojun Chang**[‡,ξ]

[†]School of Computing Technologies, RMIT University, Australia
[‡]University of Technology Sydney, [ξ]Mohamed bin Zayed University of Artificial Intelligence
`1024peng@gmail.com, andy.song@rmit.edu.au, haytham.fayek@ieee.org`
`vic.ciesielski@rmit.edu.au, xiaojun.chang@uts.edu.au`

## ABSTRACT

Training-free metrics (a.k.a. zero-cost proxies) are widely used to avoid resource-intensive neural network training, especially in Neural Architecture Search (NAS). Recent studies show that existing training-free metrics have several limitations, such as limited correlation and poor generalisation across different search spaces and tasks. Hence, we propose Sample-Wise Activation Patterns and its derivative, SWAP-Score, a novel high-performance training-free metric. It measures the expressivity of networks over a batch of input samples. The SWAP-Score is strongly correlated with ground-truth performance across various search spaces and tasks, outperforming 15 existing training-free metrics on NAS-Bench-101/201/301 and TransNAS-Bench-101. The SWAP-Score can be further enhanced by regularisation, which leads to even higher correlations in cell-based search space and enables model size control during the search. For example, Spearman's rank correlation coefficient between regularised SWAP-Score and CIFAR-100 validation accuracies on NAS-Bench-201 networks is 0.90, significantly higher than 0.80 from the second-best metric, NWOT. When integrated with an evolutionary algorithm for NAS, our SWAP-NAS achieves competitive performance on CIFAR-10 and ImageNet in approximately 6 minutes and 9 minutes of GPU time respectively.[1]

## 1 INTRODUCTION

Performance evaluation of neural networks is critical, especially in Neural Architecture Search (NAS) which aims to automatically construct high-performing neural networks for a given task. The conventional approach evaluates candidate networks by feed-forward and back-propagation training. This process typically requires every candidate to be trained on the target dataset until convergence (Liu et al., 2019; Zoph & Le, 2017), and often leads to prohibitively high computational cost (Ren et al., 2022; White et al., 2023). To mitigate this cost, several alternatives have been introduced, such as performance predictors, architecture comparators and weight-sharing strategies.

A divergent approach is the use of training-free metrics, also known as zero-cost proxies (Chen et al., 2021a; Lin et al., 2021; Lopes et al., 2021; Mellor et al., 2021; Mok et al., 2022; Tanaka et al., 2020b; Li et al., 2023). The aim is to eliminate the need for network training entirely. These metrics are either positively or negatively correlated with the networks' ground-truth performance. Typically, they only necessitate a few forward or backward passes with a mini batch of input data, making their computational costs negligible compared to traditional network performance evaluation. However, training-free metrics face several challenges: (1) Unreliable correlation with the network's ground-truth performance (Chen et al., 2021a; Mok et al., 2022); (2) Limited generalisation across different search spaces and tasks (Krishnakumar et al., 2022), even unable to consistently surpass some computationally simple counterparts like the number of network parameters or FLOPs; (3) A bias towards larger models (White et al., 2023), which means they do not naturally lead to smaller models when such models are desirable.

---

[*]Corresponding author.
[1]Our code is available at `https://github.com/pym1024/SWAP`. All experiments are conducted on a single Tesla V100 GPU.

To overcome these limitations, we introduce a novel high-performance training-free metric, Sample-Wise Activation Patterns (SWAP-Score), which is inspired by the studies of network expressivity (Montúfar et al., 2014; Xiong et al., 2020), but addresses the above limitations. It correlates with the network ground-truth performance much stronger. We rigorously evaluate its predictive capabilities across five distinct search spaces — NAS-Bench-101, NAS-Bench-201, NAS-Bench-301 and TransNAS-Bench-101-Micro/Macro, to validate whether SWAP-Score can generalise well on different types of tasks. It is benchmarked against 15 existing training-free metrics to gauge its correlation with networks' ground-truth performance. Further, the correlation of SWAP-Score can be increased by regularisation, and enables model size control during the architecture search. Finally, we demonstrate its capability by integrating SWAP-Score into NAS as a new method, SWAP-NAS. This method combines the efficiency of SWAP-Score with the effectiveness of population-based evolutionary search, which is typically computationally intensive. This work's primary contributions are as follows:

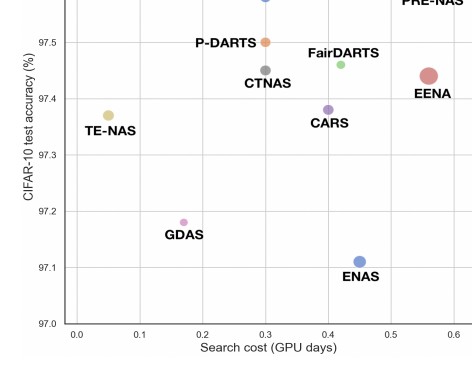

Figure 1: Search cost and performance comparison between SWAP-NAS and other SoTA NAS on CIFAR-10. Methods over 1 GPU day are not included. The dot size indicates the model size.

- We introduce **S**ample-**W**ise **A**ctivation **P**atterns and its derivative, SWAP-Score, a novel high correlation training-free metric. Unlike revealing network expressivity through standard activation patterns, SWAP-Score offers a significantly higher capability to differentiate networks. Comprehensive experiments validate its robust generalisation and superior performance across five benchmark search spaces, i.e., stack-based and cell-based, and seven tasks, i.e., image classification, object detection, autoencoding and jigsaw puzzle, outperforming 15 existing training-free metrics including recent proposed NWOT and ZiCo.

- We enable model size control in architecture search by adding regularisation to SWAP-Score. Besides, regularised SWAP-Score can more accurately align with the performance distribution of cell-based search spaces.

- We propose an ultra-fast NAS algorithm, SWAP-NAS, by integrating regularised SWAP-Score with evolutionary search. It can complete a search on CIFAR-10 in a mere **0.004 GPU days (6 minutes)**, outperforming SoTA NAS methods in both speed and performance, as illustrated in Fig. 1. A direct search on ImageNet requires only **0.006 GPU days (9 minutes)** to achieve SoTA NAS, demonstrating its high efficiency and performance.

## 2  RELATED WORK

Early network evaluation approaches often train each candidate network individually. For instance, AmoebaNet (Real et al., 2019) employs an evolutionary search algorithm and trains every sampled network from scratch, requiring approximately 3150 GPU days to search on the CIFAR-10 dataset. The resulting architecture, when transferred to the ImageNet dataset, achieves a top-1 accuracy of 74.5%. Performance predictors can reduce evaluation costs, such as training regression models based on architecture-accuracy pairs (Liu et al., 2018; Luo et al., 2020; Shi et al., 2020; Wen et al., 2020; Peng et al., 2023). Another strategy is architecture comparator, which selects the better architecture from a pair of candidates through pairwise comparison (Dudziak et al., 2020; Chen et al., 2021b). Nevertheless, both approaches necessitate the preparation of training data consisting of architecture-accuracy pairs. One alternative evaluation strategy is weight-sharing among candidate architectures, eliminating the need to train each candidate individually (Cai et al., 2018; Pham et al., 2018; Liang et al., 2019; Liu et al., 2019; Dong & Yang, 2019b; Xu et al., 2020; Chu et al., 2020). With this strategy, the computational overhead in NAS can be substantially reduced from tens of thousands of GPU hours to dozens, or less. For example, DARTS (Liu et al., 2019) combines the

one-shot model, a representative weight-sharing strategy, with a gradient-based search algorithm, requiring only 4 GPU days to achieve a test accuracy of 97.33% on CIFAR-10. Unfortunately, it is problematic to share trained weights among heterogeneous networks. In addition, weight-sharing strategies often suffer from an optimisation gap between the ground-truth performance and the approximated performance evaluated by these strategies (Shi et al., 2020; Xie et al., 2021). Further, the one-shot model, treats the entire search space as an over-parameterised super-network and that is difficult to optimise and work with limited resources.

In comparison with the above strategies, training-free metrics further reduce evaluation cost as no training is required (Tanaka et al., 2020b; Chen et al., 2021a; Lin et al., 2021; Lopes et al., 2021; Mellor et al., 2021; Mok et al., 2022; Li et al., 2023). For instance, by combining two training-free metrics, the number of linear regions (Xiong et al., 2020) and the spectrum of the neural tangent kernel (Jacot et al., 2018), TE-NAS (Chen et al., 2021a) only requires 0.05 GPU days for CIFAR-10 and 0.17 GPU days for ImageNet. NWOT (Mellor et al., 2021) explores the overlap of activations between data points in untrained networks as an indicator of performance. Zen-NAS (Lin et al., 2021) observed that most training-free NAS were inferior to the training-based state-of-the-art NAS methods. Thus, they proposed Zen-Score, a training-free metric inspired by the network expressivity studies (Jacot et al., 2018; Xiong et al., 2020). With a specialised search space, Zen-NAS achieves 83.6% top-1 accuracy on ImageNet in 0.5 GPU day, which is the first training-free NAS that outperforms training-based NAS methods. However, Zen-Score is not mathematically well-defined in irregular design spaces, thus, it cannot be applied to search spaces like cell-based. ZiCo (Li et al., 2023) noted that none of the existing training-free metrics could work consistently better than the number of network parameters. By leveraging the mean value and standard deviation of gradients across different training batches as the indicator, they proposed ZiCo, which demonstrates consistent and better performance on several search spaces and tasks, than the number of network parameters. However, it is still inferior to another naive metric, FLOPs. A recent empirical study, NAS-Bench-Suite-Zero (Krishnakumar et al., 2022), evaluates 13 training-free metrics on multiple tasks. Their results indicate that most training-free metrics do not generalise well across different types of search spaces and tasks. Moreover, simple baselines such as the number of network parameters and FLOPs show better performance than some training-free metrics which are more computationally complex.

## 3  SAMPLE-WISE ACTIVATION PATTERNS AND SWAP-SCORE

To address the aforementioned challenges, we introduce SWAP-Score. Similar to NWOT and Zen-Score, SWAP-Score is also inspired by studies on network expressivity, aiming to uncover the expressivity of deep neural networks by examining their activation patterns. What sets SWAP-Score apart from other training-free metrics is its focus on sample-wise activation patterns, which offers high correlation and robust performance across a wide range of search spaces, including both stack-based and cell-based, as well as diverse tasks, such as image classification, object detection, scene classification, autoencoding and jigsaw puzzles. The following section introduces the idea of revealing a network's expressivity by examining the standard activation patterns. Then SWAP-Score is presented which is to better measure the network expressivity through sample-wise activation patterns. Lastly, we add regularisation to the SWAP-Score.

### 3.1  STANDARD ACTIVATION PATTERNS, NETWORK'S EXPRESSIVITY & LIMITATIONS

The studies on exploring the expressivity of deep neural networks (Pascanu et al., 2013; Montúfar et al., 2014; Xiong et al., 2020) demonstrate that, networks employing piecewise linear activation functions such as ReLU (Nair & Hinton, 2010), each ReLU function partitions its input space into two regions: either zero or a non-zero positive value. These ReLU activation functions introduce piecewise linearity into the network. Since the composition of piecewise linear functions remains piecewise linear, a ReLU neural network can be viewed as a piecewise linear function. Consequently, the input space of such a network can be divided into multiple distinct segments, each referred to as a linear region. The number of distinct linear regions serves as an indicator of the network's functional complexity. A network with more linear regions is capable of capturing more complex features in the data, thereby exhibiting higher expressivity.

Following this idea, the network's expressivity can be revealed by counting the cardinality of a set composed of standard activation patterns.

**Definition 3.1.** Given $\mathcal{N}$ as a ReLU deep neural network, $\theta$ as a fixed set of network parameters (randomly initialised weights and biases) of $\mathcal{N}$, a batch of inputs containing $S$ samples, the standard activation pattern, $\mathbb{A}_{\mathcal{N},\theta}$, is defined as a set of post-activation values shown as follows:

$$\mathbb{A}_{\mathcal{N},\theta} = \left\{ \mathbf{p}^{(s)} : \mathbf{p}^{(s)} = \mathbb{1}(p_v^{(s)})_{v=1}^V, \ s \in \{1, \ldots, S\} \right\}, \tag{1}$$

where $V$ denotes the number of intermediate values feeding into ReLU layers. $p_v^{(s)}$ denotes a single post-activation value from the $v^{th}$ intermediate value at $s^{th}$ sample. $\mathbb{1}(x)$ is the indicator function that identifies the unique activation patterns. In the context of ReLU networks, the $Signum$ function can be adopted as the indicator function, that converts positive non-zero values to one while leaving zero values unchanged. Consequently, $\mathbb{A}_{\mathcal{N},\theta}$ represents a set containing the binarised post-activation values produced by network $\mathcal{N}$ with parameters $\theta$ and $S$ input samples.

The set $\mathbb{A}_{\mathcal{N},\theta}$ can also be viewed as a matrix, with each element as a row representing a vector $\mathbb{1}(p_v^{(s)})_{v=1}^V$ of binarised post-activation values over all intermediate values in $V$. Each value or cell corresponds to $\mathbb{1}(p_v^{(s)})$ as defined in Eq. 1. The upper bound of the cardinality of $\mathbb{A}_{\mathcal{N}}$ is equal to the number of input samples, $S$. Since $V$ represents the number of intermediate values feeding into the activation layers, define $\mathcal{N}$ contains $L$ layers, the dimensionality of input is $C \times W \times H$, we have:

$$V = \begin{cases} \sum_{l=1}^L n_l, & \text{if } \mathcal{N} \text{ is MLP,} \\ \sum_{l=1}^L \left( c_l \times (\lfloor \frac{w_l - k_l}{t_l} \rfloor + 1) \times (\lfloor \frac{h_l - k_l}{t_l} \rfloor + 1) \right), & \text{if } \mathcal{N} \text{ is CNN.} \end{cases} \tag{2}$$

For multi-layer perceptrons (MLP), $n_l$ denotes the number of hidden neurons in $l^{th}$ layer. For convolutional neural networks (CNN), $c_l$ denotes the number of convolution kernels, $t_l$ denotes the stride of convolution kernels, $k_l$ denotes the kernel size in $l^{th}$ layer. Hence, the length of $\mathbb{1}(p_v^{(s)})_{v=1}^V$ is influenced by the dimensionality of the input samples. Given the same number of inputs, higher-dimensional inputs or deeper networks will generate more intermediate values, making it more likely to produce distinct vectors and reach the upper bounds of cardinality. Fig. 2 illustrates two examples, where (a) shows the matrix $\mathbb{A}_{\mathcal{N},\theta}$ derived from low-dimensional inputs, while that of (b) is derived from higher-dimensional inputs. Due to pattern duplication, the cardinality in (a) is $4$, whereas in (b) it is $5$. Note, the latter reaches the upper limit, the number of input samples, $S$, that is 5 in this case. This highlights the limitation of examining standard activation patterns for measuring the network's expressivity. Methods like TE-NAS (Chen et al., 2021a) only allow inputs of small dimensions, e.g., $1 \times 3 \times 3$. Otherwise, the metric values from different networks will all approach the number of input samples, making them indistinguishable.

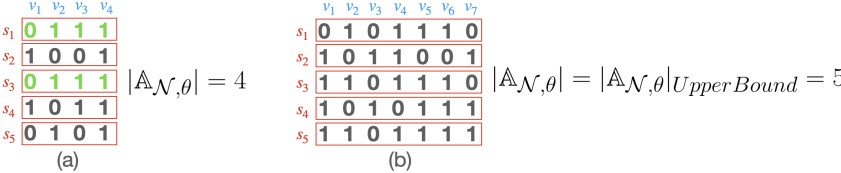

Figure 2: Two examples of $\mathbb{A}_{\mathcal{N},\theta}$ with different inputs. Green denotes duplicate patterns.

## 3.2 SAMPLE-WISE ACTIVATION PATTERNS

SWAP-Score addresses the limitation identified above. It also uses piecewise linear activation functions to measure the expressivity of deep neural networks. However, SWAP-Score does so on sample-wise activation patterns, resulting in a significantly higher upper bound, providing more space to discriminate or separate networks with different performances.

**Definition 3.2** (Sample-Wise Activation Patterns)**.** Given a ReLU deep neural network $\mathcal{N}$, $\theta$ as a fixed set of network parameters (randomly initialised weights and biases) of $\mathcal{N}$, a batch of inputs containing $S$ samples, sample-wise activation patterns $\hat{\mathbb{A}}_{\mathcal{N},\theta}$ is defined as follows:

$$\hat{\mathbb{A}}_{\mathcal{N},\theta} = \left\{ \mathbf{p}^{(v)} : \mathbf{p}^{(v)} = \mathbb{1}(p_s^{(v)})_{s=1}^S, \ v \in \{1, \ldots, V\} \right\}, \tag{3}$$

where $p_s^{(v)}$ denotes a single post-activation value from the $s^{th}$ sample at the $v^{th}$ intermediate value.

Note, in comparison with $\mathbb{A}_{\mathcal{N},\theta}$ in Eq. 1, the vectors here are now sample-wise rather than intermediate value-wise as in standard activation patterns. In sample-wise activation patterns, $\mathbb{1}(p_s^{(v)})_{s=1}^S$ is a vector containing binarised post-activation values across all samples in $S$.

**Definition 3.3** (SWAP-Score $\Psi$). Given a SWAP set $\hat{\mathbb{A}}_{\mathcal{N},\theta}$, the SWAP-Score $\Psi$ of network $\mathcal{N}$ with a fixed set of network parameters $\theta$ is defined as the cardinality of the set, computed as follows:

$$\mathbf{\Psi}_{\mathcal{N},\theta} = \left| \hat{\mathbb{A}}_{\mathcal{N},\theta} \right|. \tag{4}$$

Fig. 3 illustrates the connection and the difference between $\mathbb{A}_{\mathcal{N},\theta}$ and $\hat{\mathbb{A}}_{\mathcal{N},\theta}$ in a simplified form. Both sets are based on the same network with the same input. Hence, they have the same set of binarised post-activation values but are represented differently. The upper bound of the cardinality using standard activation patterns $\mathbb{A}_{\mathcal{N},\theta}$ is 5. In contrast, the upper bound of SWAP-Score $\Psi$ extends to 7. According to Eq. 2, the number of intermediate values $V$ grows exponentially with either an increase in the dimensionality of the input samples or the depth of the neural networks. This implies that the number of intermediate values, $V$, would be much larger than the number of input samples, $S$. As a result, SWAP has a significantly higher capacity for distinct patterns, which allows SWAP-Score to measure the network's expressivity more accurately. Specifically, this characteristic leads to a high correlation with the ground-truth performance of network $\mathcal{N}$ (see Section 4 for more details).

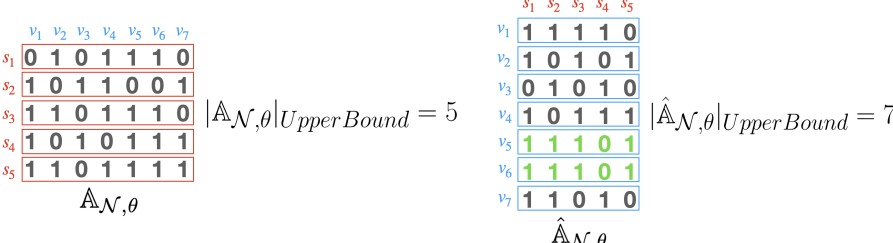

Figure 3: Illustration of $\mathbb{A}_{\mathcal{N},\theta}$ and $\hat{\mathbb{A}}_{\mathcal{N},\theta}$ from a network $\mathcal{N}$. Green denotes the duplicate patterns.

### 3.3 REGULARISATION

As mentioned earlier, training-free metrics tend to bias towards larger models (White et al., 2023), meaning they do not naturally lead to smaller models when such models are desirable. Using the convolutional neural network as an example, the convolution operations with larger kernel sizes or more channels have more parameters while producing more intermediate values compared to operations like skip connection (He et al., 2016) or pooling layer. Consequently, larger networks typically yield higher metric values, which may not always be desirable. To mitigate this bias, we add regularisation for SWAP-Score.

**Definition 3.4** (Regularisation). Given the total number of network parameters $\Theta$, coefficients $\mu$ and $\sigma$, SWAP regularisation is defined as follows:

$$f(\Theta) = e^{-\left(\frac{(\Theta-\mu)^2}{\sigma}\right)}. \tag{5}$$

**Definition 3.5** (Regularised SWAP-Score). Given regularisation function $f(\Theta)$, SWAP-Score $\Psi$ of network $\mathcal{N}$ with a fixed set of network parameters $\theta$, regularised SWAP-Score $\Psi'$ is defined as:

$$\mathbf{\Psi}'_{\mathcal{N},\theta} = \mathbf{\Psi}_{\mathcal{N},\theta} \times f(\Theta). \tag{6}$$

Regularisation function $f(\Theta)$ is a bell-shaped curve. Coefficient $\mu$ controls the centre position of this curve. Coefficient $\sigma$ adjusts the shape of the curve. By explicitly setting the values for $\mu$ and $\sigma$, the regularised SWAP-Score, $\mathbf{\Psi}'_{\mathcal{N},\theta}$, can guide the resulting architectures toward a desired range of model sizes.

## 4 EXPERIMENTS AND RESULTS

Comprehensive experiments are conducted to confirm the effectiveness of SWAP-Score. Firstly, SWAP-Score and its regularised version are benchmarked against 15 other training-free metrics across five distinct search spaces and seven tasks (Section 4.1). Subsequently, we integrate regularised SWAP-Score with evolutionary search as SWAP-NAS, to evaluate its performance in NAS. Further, state-of-the-art NAS methods are compared in terms of both search performance and efficiency (Section 4.2). Additionally, our ablation study demonstrates the effectiveness of SWAP-Scores, particularly when handling large size inputs and in model size control (Section 4.3). The tasks involved in the experiments are:

1. **Image classification tasks**: CIFAR-10 / CIFAR-100 (Krizhevsky, 2009), ImageNet-1k (Deng et al., 2009) and ImageNet16-120 (Chrabaszcz et al., 2017).

2. **Object detection task**: Taskonomy dataset (Zamir et al., 2018).

3. **Scene classification task**: MIT Places dataset (Zhou et al., 2018).

4. **Jigsaw puzzle**: the input is divided into patches and shuffled according to preset permutations. The objective is to classify which permutation is used (Krishnakumar et al., 2022).

5. **Autoencoding**: a pixel-level prediction task that encodes images into low-dimensional latent representation then reconstructs the raw image (Krishnakumar et al., 2022).

Six search spaces are used to verify the advantages of SWAP-Score and regularised SWAP-Score:

1. **NAS-Bench-101** (Ying et al., 2019): a cell-based benchmark search space which contains 423624 unique architectures trained on CIFAR-10. The architectures are designed as ResNet-like and Inception-like (He et al., 2016; Szegedy et al., 2016).

2. **NAS-Bench-201** (Dong & Yang, 2020): a cell-based benchmark search space which contains 15625 unique architectures trained on CIFAR-10, CIFAR-100 and ImageNet16-120.

3. **NAS-Bench-301** (Siems et al., 2020): a surrogate benchmark space which contains architectures sampled from the DARTS search space.

4. **TransNAS-Bench-101-Mirco/Macro** (Duan et al., 2021): consists of a micro (cell-based) search space of size 4096, and a macro (stack-based) search space of size 3256.

5. **DARTS** (Liu et al., 2019): a cell-based search space contains $10^{18}$ possible architectures.

### 4.1 SWAP-SCORES V.S. 15 OTHER TRAINING-FREE METRICS ON CORRELATION

Our SWAP-Scores are compared against 15 training-free (TF) metrics (Mellor et al., 2021; Abdelfattah et al., 2021; Lin et al., 2021; Lopes et al., 2021; Turner et al., 2020; Ning et al., 2021; Wang et al.; Lee et al.; Tanaka et al., 2020a; Li et al., 2023) across different search spaces and tasks, in terms of correlation. These extensive studies follow the same setup as NAS-Bench-Suite-Zero (Krishnakumar et al., 2022), which is a standardised framework for verifying the effectiveness of training-free metrics. The hyper-parameters, such as batch size, input data, sampled architectures and random seeds are fixed and consistently applied to all training-free metrics as NAS-Bench-Suite-Zero. The comparison is shown in Fig. 4, where each column is the Spearman coefficients of all metrics on one task with one given search space. They are computed on 1000 randomly sampled architectures. Each value in Fig. 4 is an average of five independent runs with different random seeds. The $\mu$ and $\sigma$ setups for the regularisation function are determined by the model size distribution based on 1000 randomly sampled architectures. The process only requires a few seconds for each search space.

The results in Fig. 4 clearly demonstrate the exceptional predictive capability of SWAP-Scores across diverse types of search spaces and tasks. Notably, both SWAP-Scores outperform 15 other

| | TNB101_MICRO-AUTOENC | TNB101_MACRO-OBJECT | TNB101_MACRO-JIGSAW | NB101-CF10 | TNB101_MACRO-AUTOENC | TNB101_MACRO-SCENE | NB301-CF10 | TNB101_MICRO-JIGSAW | TNB101_MICRO-OBJECT | TNB101_MICRO-SCENE | NB201-IMGNT | NB201-CF10 | NB201-CF100 |
|---|---|---|---|---|---|---|---|---|---|---|---|---|---|
| plain | 0.07 | -0.19 | -0.30 | -0.32 | -0.11 | -0.19 | -0.33 | 0.35 | 0.34 | 0.23 | -0.24 | -0.25 | -0.19 |
| grasp | -0.12 | -0.64 | -0.27 | 0.27 | -0.02 | -0.43 | 0.34 | -0.13 | -0.23 | -0.27 | 0.54 | 0.51 | 0.53 |
| num_lr | 0.00 | 0.00 | 0.00 | 0.00 | 0.00 | 0.00 | -0.02 | 0.00 | 0.00 | 0.00 | 0.07 | 0.07 | 0.07 |
| fisher | -0.58 | -0.30 | -0.26 | -0.28 | -0.18 | -0.12 | -0.27 | 0.32 | 0.46 | 0.67 | 0.47 | 0.49 | 0.53 |
| grad_norm | -0.32 | -0.55 | -0.27 | -0.24 | 0.32 | -0.34 | -0.03 | 0.37 | 0.40 | 0.66 | 0.55 | 0.57 | 0.62 |
| snip | -0.26 | -0.37 | -0.20 | -0.19 | 0.20 | -0.14 | -0.04 | 0.43 | 0.47 | 0.71 | 0.55 | 0.58 | 0.62 |
| epe_nas | 0.00 | -0.01 | 0.02 | -0.00 | 0.00 | -0.01 | 0.00 | 0.16 | 0.40 | 0.52 | 0.33 | 0.70 | 0.58 |
| l2_norm | 0.05 | 0.09 | 0.15 | 0.51 | -0.19 | 0.29 | 0.46 | 0.35 | 0.33 | 0.53 | 0.68 | 0.68 | 0.71 |
| zen | 0.14 | 0.10 | 0.24 | 0.60 | -0.01 | 0.28 | 0.44 | 0.52 | 0.55 | 0.72 | 0.38 | 0.33 | 0.34 |
| jacov | 0.18 | 0.07 | 0.19 | -0.29 | 0.46 | 0.19 | -0.05 | 0.56 | 0.51 | 0.75 | 0.70 | 0.74 | 0.70 |
| params | -0.00 | 0.17 | 0.17 | 0.38 | -0.18 | 0.33 | 0.46 | 0.44 | 0.46 | 0.63 | 0.69 | 0.72 | 0.73 |
| synflow | 0.00 | 0.12 | 0.34 | 0.31 | 0.00 | 0.28 | 0.18 | 0.48 | 0.49 | 0.72 | 0.75 | 0.73 | 0.76 |
| zico | 0.13 | 0.04 | 0.08 | 0.37 | 0.07 | 0.23 | 0.53 | 0.52 | 0.50 | 0.70 | 0.77 | 0.76 | 0.79 |
| flops | -0.01 | 0.79 | 0.64 | 0.37 | 0.76 | 0.85 | 0.43 | 0.45 | 0.46 | 0.65 | 0.67 | 0.69 | 0.71 |
| nwot | 0.03 | 0.84 | 0.76 | 0.32 | 0.66 | 0.89 | 0.48 | 0.43 | 0.40 | 0.60 | 0.77 | 0.77 | 0.80 |
| **swap** | -0.09 | 0.86 | 0.77 | 0.47 | 0.68 | 0.83 | 0.58 | 0.43 | 0.41 | 0.57 | 0.76 | 0.79 | 0.81 |
| **reg_swap** | -0.00 | 0.86 | 0.77 | 0.75 | 0.68 | 0.83 | 0.63 | 0.47 | 0.48 | 0.67 | 0.87 | 0.88 | 0.90 |

Figure 4: Spearman's rank correlation coefficients between TF-metric values and networks' ground-truth performance for 15 existing metrics and our two SWAP-Scores. The rows and columns are sorted based on mean scores of five independent experiments for each metric.

metrics in the majority of the evaluations. One interesting observation is the significant enhancement in SWAP-Score's performance when regularisation is applied (noted as 'reg_swap'), although its original intention is to control the model size during architecture search. This improvement is particularly evident in cell-based search spaces, including NAS-Bench-101, NAS-Bench-201, NAS-Bench-301, and TransNAS-Bench-101-Micro. However, it is worth noting that regularisation does not appear to impact the correlation results in stack-based space, TransNAS-Bench-101-Macro.

## 4.2 SWAP-NAS ON DARTS SPACE

To further validate the effectiveness of SWAP-Score, we utilise it for NAS by integrating the regularised version with evolutionary search as SWAP-NAS. DARTS search space is used for the following experiments, given its widespread presence in NAS studies, allowing a fair comparison with SoTA methods. For the evolutionary search, SWAP-NAS is similar to Real et al. (2019), but uses regularised SWAP-Score as the performance measure. Parent architectures generate possible offspring iteratively in each search cycle, with both mutation and crossover operations. Unlike many training-based NAS approaches that initiate the search on CIFAR-10 and later transfer the architecture to ImageNet, SWAP-NAS conducts direct searches on ImageNet. This is made feasible because of the high efficiency of SWAP-Score.

### 4.2.1 RESULTS ON CIFAR-10

Table 1 shows the results of architectures found by SWAP-NAS from DARTS space for CIFAR-10. The networks' training strategy and hyper-parameters are exactly following the setup in DARTS (Liu et al., 2019). Three variations of SWAP-NAS are presented, with different regularisation parameters $\mu$ and $\sigma$. Regardless of these parameters, SWAP-NAS only requires 0.004 GPU days, or 6 minutes. That is 6.5 times faster than the SoTA (TE-NAS). Meanwhile, the architectures found by SWAP-NAS also outperform most of the previous work. In addition, the capability of model size control is demonstrated. SWAP-NAS-A, with small $\mu$ and $\sigma$ values, generates smaller networks but also suffers a tiny performance deterioration. While SWAP-NAS-C, with large $\mu$ and $\sigma$, achieves the best error rate but at the cost of a slightly bloated network. This capability allows practitioners to find a balance between performance and model size according to the need of the task.

| | Test Error (%) | Params (M) | GPU Days | Search Method | Evaluation |
|---|---|---|---|---|---|
| PNAS (Liu et al., 2018) | 3.34±0.09 | 3.2 | 225 | SMBO | Predictor |
| EcoNAS (Zhou et al., 2020)† | 2.62±0.02 | 2.9 | 8 | Evolution | Conventional |
| DARTS (Liu et al., 2019)† | 3.00±0.14 | 3.3 | 4 | Gradient | One-shot |
| EvNAS (Sinha & Chen, 2021)† | 2.47±0.06 | 3.6 | 3.83 | Evolution | One-shot |
| RandomNAS (Li & Talwalkar, 2020)† | 2.85±0.08 | 4.3 | 2.7 | Random | One-shot |
| EENA (Zhu et al., 2019) | 2.56⋆ | 8.47 | 0.65 | Evolution | Weights Inherit |
| PRE-NAS (Peng et al., 2023)† | 2.49±0.09 | 4.5 | 0.6 | Evolution | Predictor |
| ENAS (Pham et al., 2018) | 2.89⋆ | 4.6 | 0.45 | Reinforce | One-shot |
| FairDARTS (Chu et al., 2020)† | 2.54⋆ | 2.8 | 0.42 | Gradient | One-shot |
| CARS (Yang et al., 2020)† | 2.62⋆ | 3.6 | 0.4 | Evolution | One-shot |
| P-DARTS (Chen et al., 2019)† | 2.50⋆ | 3.4 | 0.3 | Gradient | One-shot |
| TNASP (Lu et al., 2021)† | 2.57±0.04 | 3.6 | 0.3 | Evolution | Predictor |
| PINAT (Lu et al., 2023)† | 2.54±0.08 | 3.6 | 0.3 | Evolution | Predictor |
| GDAS (Dong & Yang, 2019a) | 2.82⋆ | 2.5 | 0.17 | Gradient | One-shot |
| CTNAS (Chen et al., 2021b) | 2.59±0.04 | 3.6 | 0.1+0.3 | Reinforce | Predictor |
| TE-NAS (Chen et al., 2021a)† | 2.63±0.064 | 3.8 | 0.03 | Pruning | Training-free |
| **SWAP-NAS-A ($\mu$=0.9, $\sigma$=0.9)** † | 2.65±0.04 | 3.06 | 0.004 | Evolution | Training-free |
| **SWAP-NAS-B ($\mu$=1.2, $\sigma$=1.2)** † | 2.54±0.07 | 3.48 | 0.004 | Evolution | Training-free |
| **SWAP-NAS-C ($\mu$=1.5, $\sigma$=1.5)** † | 2.48±0.09 | 4.3 | 0.004 | Evolution | Training-free |

Table 1: Performance comparison between the networks found by SWAP-NAS and other methods on CIFAR-10. A lower test error rate is better. † means the method adopted DARTS space. ⋆ indicates the original paper only reported their best result.

| | Test Error Top1/Top5 | Params (M) | GPU Days | Search Method | Evaluation | Searched Dataset |
|---|---|---|---|---|---|---|
| ProxylessNAS (Cai et al., 2018) | 24.9 / 7.5 | 7.1 | 8.3 | Gradient | One-shot | ImageNet |
| DARTS (Liu et al., 2019)† | 26.7 / 8.7 | 4.7 | 4 | Gradient | One-shot | CIFAR-10 |
| EvNAS (Sinha & Chen, 2021)† | 24.4 / 7.4 | 5.1 | 3.83 | Evolution | One-shot | CIFAR-10 |
| PRE-NAS (Peng et al., 2023)† | 24.0 / 7.8 | 6.2 | 0.6 | Evolution | Predictor | CIFAR-10 |
| FairDARTS (Chu et al., 2020)† | 26.3 / 8.3 | 2.8 | 0.42 | Gradient | One-shot | CIFAR-10 |
| CARS (Yang et al., 2020)† | 24.8 / 7.5 | 5.1 | 0.4 | Evolution | One-shot | CIFAR-10 |
| P-DARTS (Chen et al., 2019)† | 24.4 / 7.4 | 4.9 | 0.3 | Gradient | One-shot | CIFAR-10 |
| CTNAS (Chen et al., 2021b) | 22.7 / 7.5 | - | 0.1+50 | Reinforce | Predictor | ImageNet |
| ZiCo (Li et al., 2023) | 21.9 / - | - | 0.4 | Evolution | Training-free | ImageNet |
| PINAT-T (Lu et al., 2023)† | 24.9 / 7.5 | 5.2 | 0.3 | Evolution | Predictor | CIFAR-10 |
| GDAS (Dong & Yang, 2019a) | 27.5 / 9.1 | 4.4 | 0.17 | Gradient | One-shot | CIFAR-10 |
| TE-NAS (Chen et al., 2021a)† | 24.5 / 7.5 | 5.4 | 0.17 | Pruning | Training-free | ImageNet |
| QE-NAS (Sun et al., 2022)† | 25.5 / - | 3.2 | 0.02 | Evolution | Training-free | ImageNet |
| **SWAP-NAS ($\mu$=25, $\sigma$=25)** † | 24.0 / 7.6 | 5.8 | 0.006 | Evolution | Training-free | ImageNet |

Table 2: Performance comparison between the networks found by SWAP-NAS and other methods on ImageNet. A lower test error rate is better. † means the method adopted DARTS space.

### 4.2.2 RESULTS ON IMAGENET

Table 2 shows the NAS results on ImageNet, where training strategy and hyper-parameters setting are also the same in DARTS (Liu et al., 2019). The search cost of SWAP-NAS here slightly increased to 0.006 GPU days, or 9 minutes. That is still 2.3 times faster than the SoTA (QE-NAS) yet with a better performance.

### 4.3 ABLATION STUDY

The first ablation study is to further elucidate the limitation of standard activation patterns as discussed in Section 3.1. A mini batch of 32 images is provided to compute the metric values using the standard activation patterns, the sample-wise patterns, and the regularised patterns. these architectures using a mini batch of inputs. The mini batch size aligns with that in NAS-Bench-Suite-Zero (Krishnakumar et al., 2022) and other training-free metrics studies such as TE-NAS (Chen et al., 2021a). Table 3 shows results under three input sizes, $3 \times 3$, $15 \times 15$ and $32 \times 32$, the latter being the original size of CIFAR-10 images. The mean and standard deviation for each metric are calculated based on their values across 1000 architectures. The corresponding Spearman correlations to true performance are also listed. With the standard activation patterns, $|\mathbb{A}_{\mathcal{N},\theta}|$ shows tiny variation under input $3 \times 3$ and zero variation under larger size inputs, indicating its limited capability on distinguishing the differences between architectures, particularly when the input dimensionality is high. Additionally, the mean value approaches its theoretical upper bound, the number of input samples, 32. This phenomenon confirms our discussion in Section 3.1. In contrast, both SWAP-Score $\Psi_{\mathcal{N},\theta}$

and regularised SWAP-Score $\Psi'_{\mathcal{N},\theta}$ show significantly higher mean values and much more variation across the 1000 architectures. This indicates they have higher upper bounds and better capabilities to differentiate architectures. Notably, the regularised SWAP-Score exhibits even greater diversity and high correlation with the increase in input size. $\Psi'_{\mathcal{N},\theta}$ starts from 0.89 and reaches 0.93.

| | Input size $3 \times 3$ | | Input size $15 \times 15$ | | Input size $32 \times 32$ | |
|---|---|---|---|---|---|---|
| | **Metric values** | **Correlation** | **Metric values** | **Correlation** | **Metric values** | **Correlation** |
| $\lvert \mathbb{A}_{\mathcal{N},\theta} \rvert$ | $15.90 \pm 1.42$ | 0.86 | $31.0 \pm 0.0$ | 0.45 | $32.0 \pm 0.0$ | 0.34 |
| $\Psi_{\mathcal{N},\theta}$ | $42.94 \pm 10.76$ | 0.86 | $859.73 \pm 194.33$ | 0.84 | $3567.02 \pm 775.45$ | 0.90 |
| $\Psi'_{\mathcal{N},\theta}$ | $37.77 \pm 11.13$ | 0.89 | $831.31 \pm 216.11$ | 0.92 | $3412.49 \pm 867.31$ | 0.93 |

Table 3: Metric values and correlations from standard activation patterns, sample-wised patterns and regularised patterns with three input sizes (means and standard deviations from 1000 architectures).

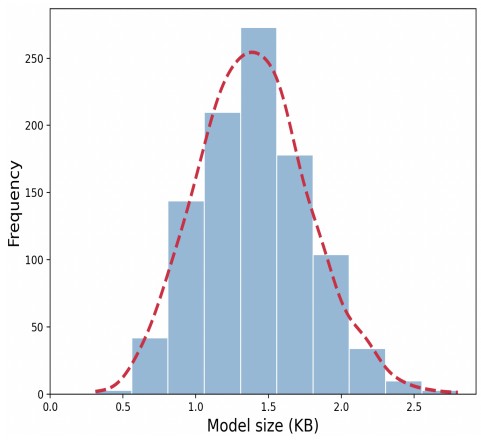

(a) Model size distribution of 1000 cell networks sampled from DARTS for CIFAR-10.

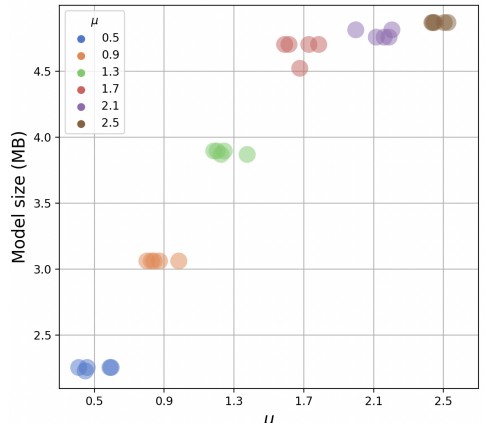

(b) Illustration of the model size control of SWAP-NAS on CIFAR-10.

Figure 5: Illustration of regularised SWAP-Score's capability on model size control in NAS.

The second ablation study illustrates regularised SWAP-Score for model size control. Fig. 5a shows the size distribution of 1000 models generated for CIFAR-10 networks from DARTS space. It is almost Gaussian, ranging from 0.5 KB to 2.5 KB. To simplify the illustration, we set $\mu = \sigma$ and increase them simultaneously from 0.5 to 2.5, with a step of 0.4. Fig. 5b visualises the relation between $\mu$, $\sigma$, and the size of models that are found by SWAP-NAS. At each $\mu$ value, SWAP-NAS runs 5 times. Random jitter is introduced in the drawing here to reduce overlaps between dots of the same $\mu$. From the figure, it can be clearly seen that $\mu$ values nicely correlate to the model sizes. The same $\mu$ value leads to almost the same model size. By adjusting $\mu$ along with $\sigma$, one can control the size of the generated model.

## 5 CONCLUSION AND FUTURE WORK

In this paper, we introduce Sample-Wise Activation Patterns and its derivative, SWAP-Score, a novel training-free network evaluation metric. The proposed SWAP-Score and its regularised version show much stronger correlations with ground-truth performance than 15 existing training-free metrics on different spaces, stack-based and cell-based, for different tasks, i.e. image classification, object detection, autoencoding, and jigsaw puzzle. In addition, the regularised SWAP-Score can enable model size control during search and can further improve correlation in cell-based search spaces. When integrated with an evolutionary search algorithm as SWAP-NAS, a combination of ultra-fast architecture search and highly competitive performance can be achieved on both CIFAR-10 and ImageNet, outperforming SoTA NAS methods. Our future work will extend the concept of SWAP-Score to other activation functions, including other piecewise linear and non-linear types like GELU.

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

# A    MODEL SIZE CONTROL DURING ARCHITECTURE SEARCH

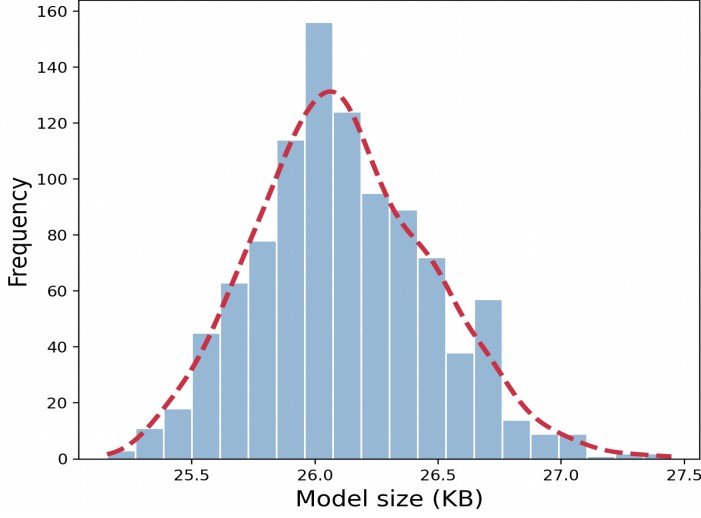

Figure 6: Illustration of the curves of regularisation function $f(\Theta)$ with different $\mu$ and $\sigma$.

As elucidated in Section 3.2, the regularization function $f(\Theta)$, defined in Equation 5, is influenced by two key parameters: $\mu$ and $\sigma$. These parameters shape the curve depicted in Fig. 6, where $\mu$ determines the curve's central position and $\sigma$ modulates its shape. Theoretically, models with a $\Theta$ value proximate to $\mu$ will have their $\Psi'_{\mathcal{N},\theta}$ values largely preserved. Conversely, a significant deviation from $\mu$ will result in a substantial attenuation of $\Psi'_{\mathcal{N},\theta}$. A smaller value of $\sigma$ sharpens the curve, thereby amplifying the regularization effect on models whose $\Theta$ values are distant from $\mu$. This leads to two results: (1) it enables control on the model size during the architecture search, and (2) it enhances the correlation of SWAP-Score in cell-based search spaces. While this section primarily focuses on the first point, the impact of varying $\mu$ and $\sigma$ on correlation is detailed in **Appendix B**.

Similar to the model size control on CIFAR-10 (Section 4.3), we can also see the size control capability of regularised SWAP-Score through SWAP-NAS on ImageNet. Fig. 7 shows the size distribution of cell networks searched from DARTS space for ImageNet. These networks are in the range of 25 KB to 27.5 KB. We still set $\mu$ and $\sigma$ to identical values and gradually increase them from 25 to 27.5 with an interval of 0.5. Fig. 8 is the relation between $\mu$ and the size of these fully stacked ImageNet networks found by SWAP-NAS with that value. Similar to the previous experiments on CIFAR-10, each $\mu$ value repeats the search 5 times. Random jitter is also introduced here. But there are much higher noticeable variations in size at each $\mu$ compared to that for CIFAR-10 (Fig. 5 (b)). Nevertheless, it can still be shown that in general, model size decreases with a reduced $\mu$. Adjusting $\mu$ would have a direct impact on the size of the generated model, even for complex tasks like ImageNet.

Figure 7: Model size distribution of 1000 cell networks sampled from DARTS space for ImageNet.

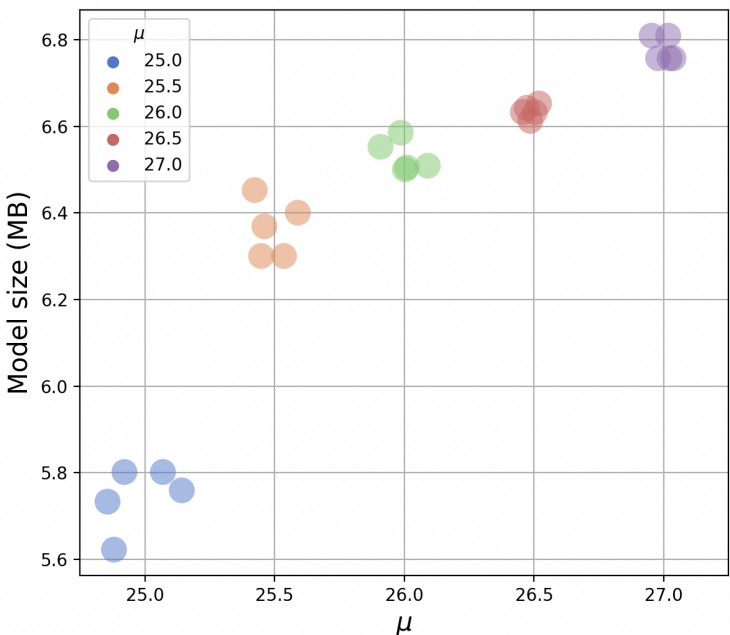

Figure 8: Illustration of the model size control of SWAP-NAS on ImageNet. The architecture search repeats 5 times at each $\mu$ value. Random jitter is used to reduce overlaps.

## B    IMPACT OF VARYING $\mu$ AND $\sigma$ TO THE CORRELATION

Firstly we demonstrate the impact of different $\mu$ and $\sigma$ on NAS-Bench-101 space (Ying et al., 2019). Following a similar procedure as in NAS-Bench-Suite-Zero (Krishnakumar et al., 2022), we utilize 6 different random seeds to form 6 groups, with each group comprising 1000 architectures randomly sampled from the NAS-Bench-101 space. One of the groups (Group 0) is used to approximate the distribution of model sizes in the NAS-Bench-101 space, not participating in the subsequent experiments. The distribution histogram obtained from Group 0 is shown in Fig. 9. The range of model size in this group is 0.3 to 31 megabytes (MB). Most of the models are in 0.3 MB to 5 MB intervals. Leveraging this information, we can better see the impact of $\mu$ and $\sigma$ on the other five groups of sampled architectures (Table 4).

Table 4 shows different combinations of $\mu$ and $\sigma$ values, and the corresponding Spearman's Rank Correlation Coefficient between regularised SWAP-Score and the ground-truth performance of the networks for the five groups, $Group$ 1 to $Group$ 5. There are four blocks in the table. The first block contains only one row, which shows the results without regularisation, in other words, the results from SWAP-Score. The second block shows the results of assigning identical values to $\mu$ and $\sigma$, ranging from 0.3 to 40, the same range of model size in MB, shown in Fig. 9. The third block shows the results of varying $\sigma$ while fixing $\mu$. The fixed $\mu$ value, 40, is chosen because it leads to the highest correlation in the second block, where $\mu = \sigma$. The last block in Table 4 shows the results of adjusting $\mu$ while fixing $\sigma$. The fixed $\sigma$ value, 30, is chosen because it gets the highest correlation in the third block.

From the results shown in Table 4, we can see that the correlation between regularised SWAP-Score and ground-truth performance is mainly affected by the value of $\mu$, as only minor changes occur in the correlations from $\sigma = 30$ to $\sigma = 10$, when we fix $\mu$ at 40 in the third block. On the contrary, reducing $\mu$ leads to a significant drop in correlation in the last block. This observation is consistent across all five groups. It is explainable as $\mu$ defines the centre position of the regularisation curve and directly determines how the regularisation curve covers the size distribution. Having said that, $\sigma$ is not insignificant. A poor choice of $\sigma$, e.g. $\mu = 40, \sigma = 0.3$, can lead to bad correlations ($< 0.05$ in Table 4). The impact of $\sigma$ on search is in a different way. A small $\sigma$ leads to a sharp curve, which narrows down the coverage of the regularisation function to a small area, meaning architectures

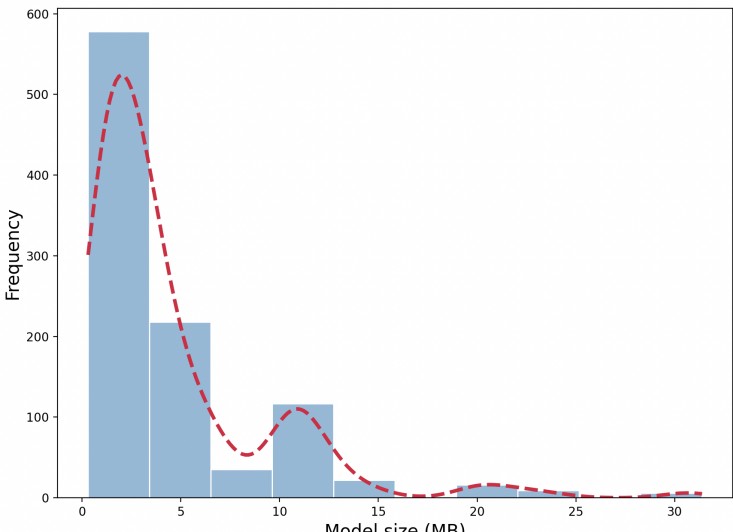

Figure 9: Histogram of model sizes in NAS-Bench-101 space, based on 1000 CIFAR-10 networks sampled in one of the six groups.

|  | $\mu$ | $\sigma$ | Group 1 | Group 2 | Group 3 | Group 4 | Group 5 |
|---|---|---|---|---|---|---|---|
| No regularisation | N/A | N/A | 0.46 | 0.49 | 0.45 | 0.45 | 0.44 |
| $\mu = \sigma$ | 0.3 | 0.3 | -0.77 | -0.78 | -0.77 | -0.75 | -0.75 |
|  | 5 | 5 | 0.03 | 0.08 | 0.07 | 0.05 | 0.04 |
|  | 10 | 10 | 0.61 | 0.63 | 0.64 | 0.6 | 0.64 |
|  | 15 | 15 | 0.74 | 0.75 | 0.73 | 0.73 | 0.73 |
|  | 20 | 20 | 0.72 | 0.75 | 0.75 | 0.74 | 0.73 |
|  | 25 | 25 | 0.76 | 0.76 | 0.76 | 0.75 | 0.74 |
|  | 30 | 30 | 0.76 | 0.77 | 0.76 | 0.74 | 0.74 |
|  | 35 | 35 | 0.76 | 0.76 | 0.76 | 0.75 | 0.74 |
|  | **40** | **40** | **0.76** | **0.76** | **0.77** | **0.75** | **0.74** |
| Varying $\sigma$ with a fixed $\mu$ | 40 | 30 | 0.76 | 0.76 | 0.76 | 0.74 | 0.75 |
|  | 40 | 20 | 0.76 | 0.76 | 0.76 | 0.74 | 0.74 |
|  | 40 | 10 | 0.75 | 0.74 | 0.73 | 0.73 | 0.73 |
|  | 40 | 0.3 | 0.04 | 0.04 | 0.03 | 0.03 | 0.03 |
| Varying $\mu$ with a fixed $\sigma$ | 20 | 30 | 0.75 | 0.76 | 0.74 | 0.73 | 0.74 |
|  | 10 | 30 | 0.64 | 0.64 | 0.63 | 0.63 | 0.66 |
|  | 5 | 30 | 0.04 | 0.08 | 0.06 | 0.04 | 0.06 |
|  | 0.3 | 30 | -0.77 | -0.78 | -0.77 | -0.74 | -0.75 |

Table 4: Impact of $\mu$ and $\sigma$ illustrated on NAS-Bench-101 architectures. The values under Group 1-5 represent the Spearman's Rank Correlation Coefficients between SWAP-Score and the ground-truth performance of these networks. Each group contains 1000 architectures sampled from NAS-Bench-101 space by different random seeds. "N/A" indicates correlations without regularisation.

outside of that size range will be heavily penalised, as their regularised SWAP-Scores after applying the regularisation function will be very low. Based on the study, our recommendation on choosing $\mu$ and $\sigma$ is that $\mu$ and $\sigma$ can be set large when the target is finding top-performing architectures.

Following the above study on NAS-Bench-101, we secondly demonstrate the impact of different $\mu$ and $\sigma$ on NAS-Bench-201 space. Six groups of architectures are sampled. One of them is used to approximate the distribution of model sizes in the NAS-Bench-201 space. The corresponding histogram is shown in Fig. 10. As can be seen in this figure, the range of model size here is 0.1 to 1.5 megabytes (MB). Accordingly, the $\mu$ values in Table 5 are in the range of 0.1 to 1.5.

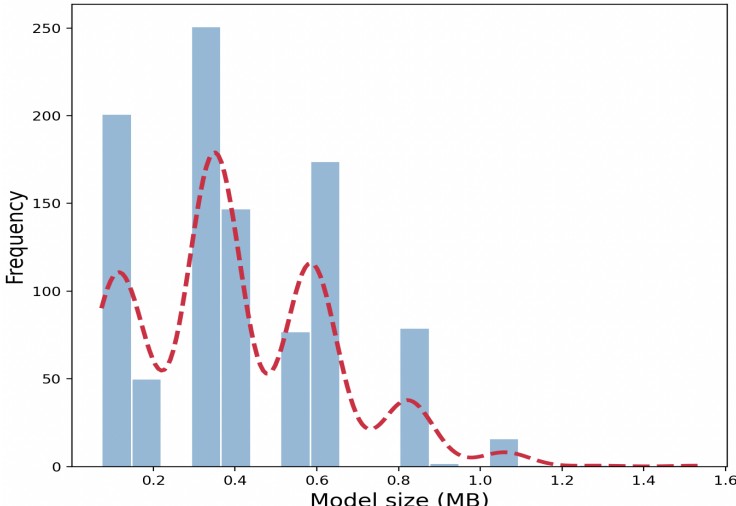

Figure 10: Histogram of model sizes in NAS-Bench-201 space, based on 1000 CIFAR-10 networks sampled in one of the six groups.

| | $\mu$ | $\sigma$ | Group 1 | Group 2 | Group 3 | Group 4 | Group 5 |
|---|---|---|---|---|---|---|---|
| No regularisation | N/A | N/A | 0.81 | 0.79 | 0.75 | 0.77 | 0.75 |
| | 0.1 | 0.1 | 0.05 | 0.04 | 0.03 | 0.03 | 0.03 |
| $\mu = \sigma$ | 0.7 | 0.7 | 0.83 | 0.82 | 0.82 | 0.84 | 0.81 |
| | **1.5** | **1.5** | **0.89** | **0.87** | **0.87** | **0.89** | **0.87** |
| Varying $\sigma$ | 1.5 | 0.87 | 0.84 | 0.86 | 0.86 | 0.84 | 0.74 |
| with a fixed $\mu$ | 1.5 | 0.4 | 0.83 | 0.78 | 0.83 | 0.79 | 0.79 |
| | 1.5 | 0.1 | 0.54 | 0.54 | 0.53 | 0.54 | 0.51 |
| Varying $\mu$ | 0.6 | 0.7 | 0.82 | 0.82 | 0.82 | 0.83 | 0.81 |
| with a fixed $\sigma$ | 0.3 | 0.7 | 0.51 | 0.49 | 0.51 | 0.54 | 0.5 |
| | 0.1 | 0.7 | 0.23 | 0.21 | 0.23 | 0.24 | 0.21 |

Table 5: Impact of $\mu$ and $\sigma$ illustrated on NAS-Bench-201 architectures. The values under Group 1-5 represent Spearman's Rank Correlation Coefficients between SWAP-Score and the ground-truth performance of these networks. Each group contains 1000 architectures sampled from NAS-Bench-201 space by different random seeds. "N/A" indicates correlations without regularisation.

Similar to Table 4, Table 5 also has four blocks, showing four scenarios of study on $\mu$ and $\sigma$. Fewer combinations are presented as the general trend here is the same as that appears in NAS-Bench-101.

The third part of the study in Section B is the impact of different $\mu$ and $\sigma$ on NAS-Bench-301 space. Similar to the previous two parts, six groups of architectures are sampled, with one for showing the distribution of model sizes in the NAS-Bench-301 space. Fig. 11 is the distribution histogram, where the size range can be seen as 1.0 to 1.8 megabytes (MB). With the same style as in Table 4 and Table 5, Table 6 lists the results of different $\mu$ and $\sigma$ in four blocks. Again, the general trend here is the same as that in NAS-Bench-101 and NAS-Bench-201.

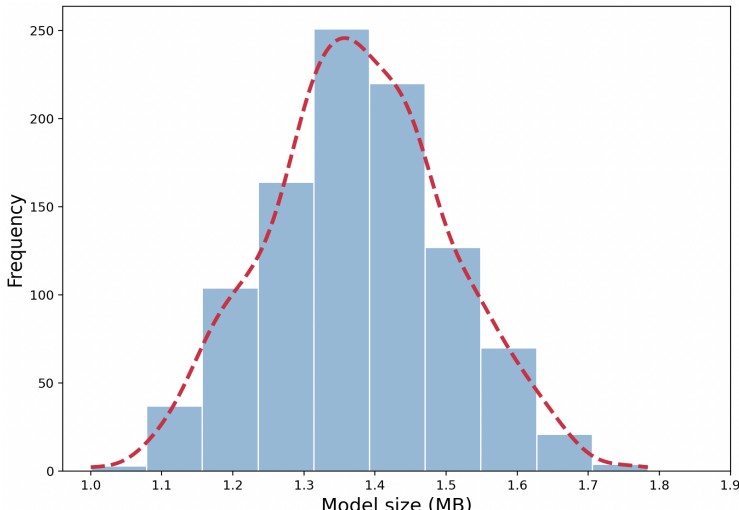

Figure 11: Histogram of model sizes in NAS-Bench-301 space, based on 1000 CIFAR-10 networks sampled in one of the six groups.

| | $\mu$ | $\sigma$ | Group 1 | Group 2 | Group 3 | Group 4 | Group 5 |
|---|---|---|---|---|---|---|---|
| No regularisation | N/A | N/A | 0.57 | 0.58 | 0.57 | 0.57 | 0.54 |
| | 1.0 | 1.0 | 0.57 | 0.58 | 0.57 | 0.57 | 0.55 |
| $\mu = \sigma$ | 1.5 | 1.5 | 0.63 | 0.61 | 0.61 | 0.61 | 0.60 |
| | **1.8** | **1.8** | **0.63** | **0.63** | **0.63** | **0.63** | **0.62** |
| Varying $\sigma$ | 1.8 | 1.5 | 0.63 | 0.63 | 0.62 | 0.63 | 0.62 |
| with a fixed $\mu$ | 1.8 | 1.3 | 0.62 | 0.62 | 0.61 | 0.61 | 0.62 |
| | 1.8 | 1.0 | 0.62 | 0.60 | 0.60 | 0.61 | 0.60 |
| Varying $\mu$ | 1.3 | 1.5 | 0.62 | 0.62 | 0.60 | 0.60 | 0.60 |
| with a fixed $\sigma$ | 1.2 | 1.5 | 0.61 | 0.61 | 0.61 | 0.59 | 0.60 |
| | 1.0 | 1.5 | 0.60 | 0.59 | 0.59 | 0.59 | 0.59 |

Table 6: Impact of $\mu$ and $\sigma$ illustrated on NAS-Bench-301 architectures. The values under Group 1-5 represent Spearman's Rank Correlation Coefficients between SWAP-Score and the ground-truth performance of these networks. Each group contains 1000 architectures sampled from NAS-Bench-301 space by different random seeds. "N/A" indicates correlations without regularisation.

## C  EVOLUTIONARY SEARCH ALGORITHM

In this section of the Appendix, we elaborate on the evolutionary search algorithm employed in SWAP-NAS. SWAP-NAS adopts cell-based search space, similar to DARTS-related works, such as Chen et al. (2019); Chu et al. (2020); Sinha & Chen (2021); Cai et al. (2018); Chen et al. (2021a); Sun et al. (2022). In terms of the search algorithm, SWAP-NAS uses evolution-based search where each step of the search is performed on a population of candidate networks rather than an individual network. This population-based approach allows for broader coverage of the search space, thereby increasing the likelihood of finding high-performance architectures. While evolutionary search algorithms are generally resource-intensive due to the need for multiple evaluations, SWAP-NAS mitigates this drawback by capitalizing on the low computational cost of SWAP-Score. As a result, we aim to deliver a NAS algorithm that is both efficient and effective, achieving high accuracy without incurring prohibitive computational costs.

The details are presented in two sections. The first section, C.1, explains the cell-based search space, while the evolutionary aspect is explained in the second section, C.2.

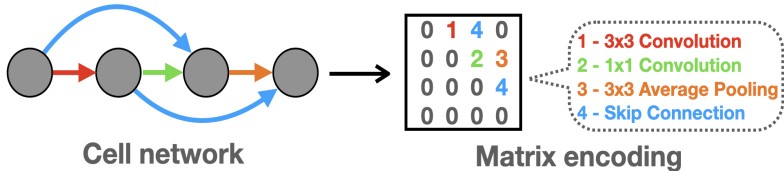

Figure 12: Illustration of a cell network and its matrix encoding. Grey circles represent the nodes inside the cell. Arrow lines indicate the flow while network operations are in different colours.

## C.1 ARCHITECTURE ENCODING OF CELL-BASED SEARCH SPACE

Fig. 12 illustrates the cell-based network representation (Liu et al., 2019; Shi et al., 2020; Zoph et al., 2018). This cell network is widely used in NAS studies (Dong & Yang, 2020; Liu et al., 2019; Siems et al., 2020; Ying et al., 2019). With this representation, the search algorithm only needs to focus on finding a good micro-structure for one cell, which is a shallow network. The final model after the search can be easily reconstructed by stacking this cell network together repetitively. The depth of the stack is determined by the difficulty of the task. As shown on the right of Fig. 12, a cell is encoded as an adjacency matrix, on which each number represents a type of connection, 1 for a $3 \times 3$ convolution, 2 is a $1 \times 1$ convolution, 3 is a $3 \times 3$ average pooling, 4 for a skip connection. The matrix in the figure is $4 \times 4$ since the example cell network has four nodes. A zero in the matrix means no connection or is not applicable. This matrix is an upper triangular matrix as it represents the directed acyclic graph (DAG). With this matrix representation, the subsequent evolutionary search can be conveniently performed by simply manipulating the matrix, for example alternating the type of connection by changing the number at the particular entry or connecting to a different node by shifting the position of the number that represents this connection.

---

**Algorithm 1** SWAP-NAS

---

**Require:** Population size $P$, Search cycle $C$, Sample size $S$, TF-metric SWAP, $mutation\_times$
1: $population \leftarrow \emptyset$
2: **while** $population < P$ **do**
3:      $model.arch \leftarrow RandomGenerateNetworks()$
4:      $model.score \leftarrow \text{SWAP}(model.arch)$
5:      Add $model$ to the $population$
6: **end while**
7: **for** $c = 1, 2, ..., C$ **do**
8:      $candidates \leftarrow S$ random samples of $population$
9:      $parent \leftarrow$ best in $candidates$ **OR** best from crossover between the best and the second best in $candidates$
10:      **while** $mutation\_times$ not reached **do**
11:          $child \leftarrow Mutate(parent)$
12:          $child.score \leftarrow \text{SWAP}(child.arch)$
13:      **end while**
14:      Add the best $child$ to the $population$
15:      Remove the worst from the $population$
16: **end for**

---

## C.2 EVOLUTIONARY SEARCH IN SWAP-NAS

As an effective search paradigm, evolutionary search is utilised in a large number of NAS methods, showing good search performance and flexibility (Lu et al., 2020; Peng et al., 2023; Real et al., 2017; 2019; Yang et al., 2020). For this very reason, SWAP-NAS is also based on an evolutionary search, but SWAP-Score is not restricted to a certain type of search algorithm. Algorithm 1 is the detailed steps in SWAP-NAS. The evolutionary search component of SWAP-NAS is slightly different from the existing evolutionary search methods used for NAS. The key distinctions are the way of producing offspring networks and how the population is updated after each search cycle. In SWAP-NAS, a tournament-style strategy is used to sample offspring networks. Half of the networks

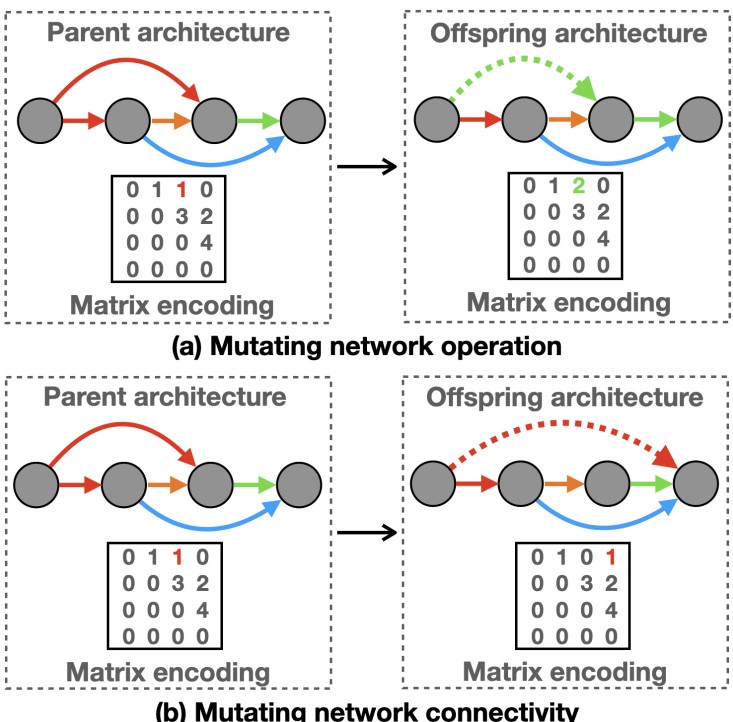

Figure 13: Illustration of the mutation operators. (a) demonstrates offspring being produced by mutating the operation of the parent architecture. (b) demonstrates offspring being produced by mutating the connectivity of the parent architecture.

are randomly selected from the population during each search cycle (Step 8). Then, in a random fashion, SWAP-NAS decides whether to perform the crossover operation on the selected network or directly use the selected network as the parent (Step 9). Therefore the *parent* will be either the best individual from the sampled networks or a network produced by the crossover between the best and the second best networks (Step 9).

In SWAP-NAS, the majority of offspring networks are generated by mutation (Step 11). There are two types of mutation, operation mutation, and connectivity mutation. They are illustrated in Fig. 13 respectively. Mutating operation is simply changing a number in the cell network matrix, as shown in Fig. 13 (a), changing the connection from node 1 to node 3 from $3 \times 3$ convolution (type number 1, red) to $1 \times 1$ convolution (type number 2, green). Mutating connectivity is shifting a connection to a different position, as shown in Fig. 13 (b), moving the $1 \times 1$ convolution connection from node 1 to node 3 to node 1 to node 4. By randomly performing these two types of mutation, SWAP-NAS can stochastically explore possible new architectures from a given parent during each search cycle.

As mentioned early, SWAP-NAS does incorporate crossover, which is a key operation, other than mutation, in evolutionary search. The use of crossover here is to avoid the search being trapped in a local optimal. The application of crossover is random as shown in Step 9 of Algorithm 1. Fig. 14 illustrates the crossover operation which is slightly different from the crossover often seen in evolutionary search. The crossover exchanges "genetic materials" between the two selected networks, e.g. some entries of the two matrices. In the example of Fig. 14, crossover swaps the incoming connections to node 4 between the two networks, e.g. exchanging the entries in $(1, 3)$ and $(2, 3)$. The two newly generated offspring networks will be evaluated using SWAP-Score. The one that scored higher will become the new parent for mutation.

Note SWAP-Score is utilised for evaluation in three places. Other than the aforementioned evaluation during the crossover, it appears in Step 4 and Step 11 in Algorithm 1, for evaluating the initial population and the new network generated by mutation. SWAP-Score is applied to the architecture,

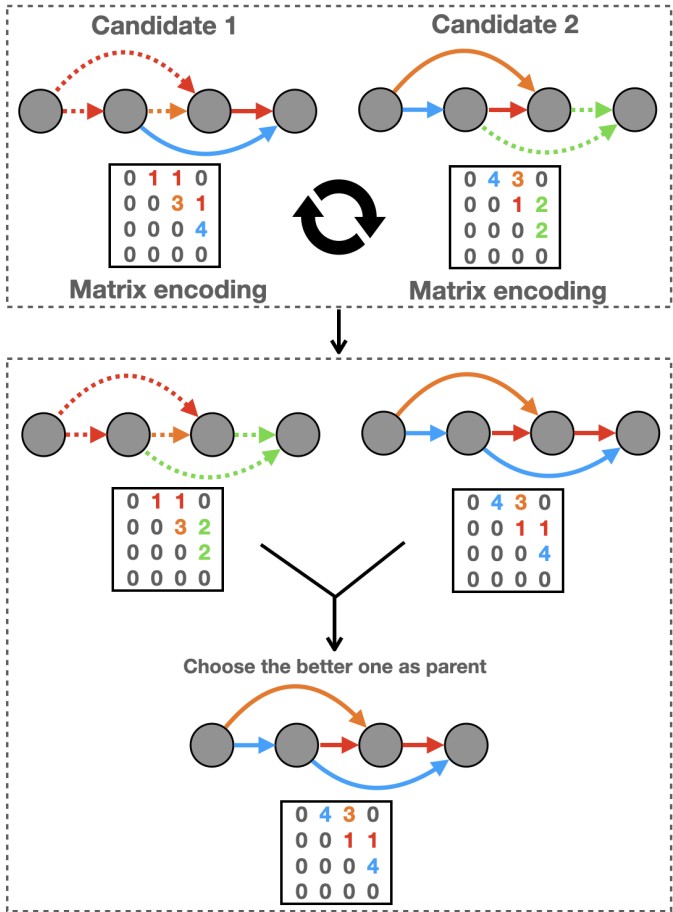

Figure 14: Illustration of the crossover operator. Candidates 1 & 2 are the best and the second-best networks from the sample. Two networks are generated by swapping components of the candidates. The better one will become the parent network for the subsequent mutation.

e.g. $model.arch$ or $child.arch$. The generated score is saved as a property of the network, e.g. $model.score$ or $child.score$.

Steps 14 & 15 of Algorithm 1 are the population updating mechanism of SWAP-NAS. Unlike the aging evolution in AmoebaNet (Real et al., 2019) which removes the oldest individual from the population, SWAP-NAS removes the worst. Theoretically, aging evolution could lead to higher diversity and better exploration of the search space. However, the elitism approach can converge faster, hence reducing the computational cost on the search algorithm side.

## D    CORRELATION OF METRICS BY INPUTS OF DIFFERENT DIMENSIONS

The correlation between these metrics and the validation accuracies of the networks is measured by Spearman's rank correlation coefficient. The full visualisation of results is shown in Fig. 15. The metric based on standard activation patterns drops dramatically when the input dimension increases. This aligns with the observation from Table 3. Both SWAP-Score and regularised SWAP-Score show a strong and consistent correlation with rising input dimensions. In particular, regularised SWAP-Score outperforms the other two, regardless of the input size. When using the original dimension of CIFAR-10, $32 \times 32$, regularised SWAP-Score shows a strong correlation, 0.93.

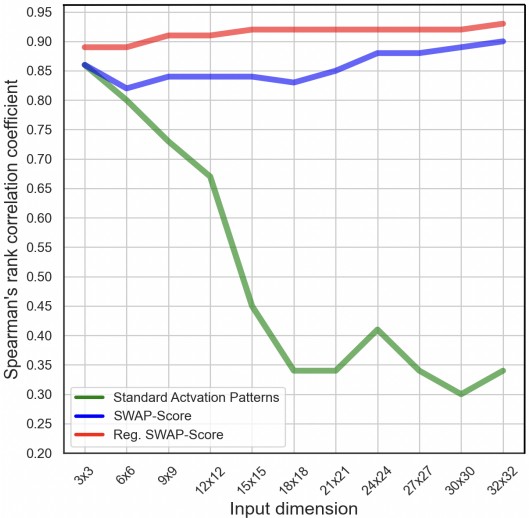

Figure 15: Spearman's coefficient between three metrics and the validation accuracies.

## E    VISUALISATION OF CORRELATION BETWEEN SWAP-SCORES AND NETWORKS' GROUND-TRUTH PERFORMANCE

Figures 16 and 17 demonstrate the correlation between the SWAP-Score/Regularised SWAP-Score and the ground-truth performance of networks across various search spaces and tasks. In these figures, each dot represents a distinct neural network. The visualisations effectively demonstrate a strong correlation between the SWAP-Score and the ground-truth performance across the majority of search spaces and tasks. Furthermore, the application of the regularisation function results in a more concentrated distribution of dots, indicating an enhanced correlation with the networks' ground-truth performance.

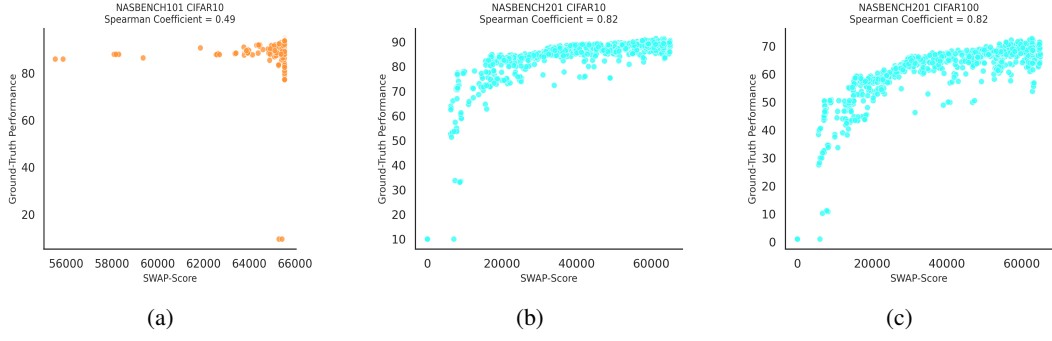

(a)                                    (b)                                    (c)

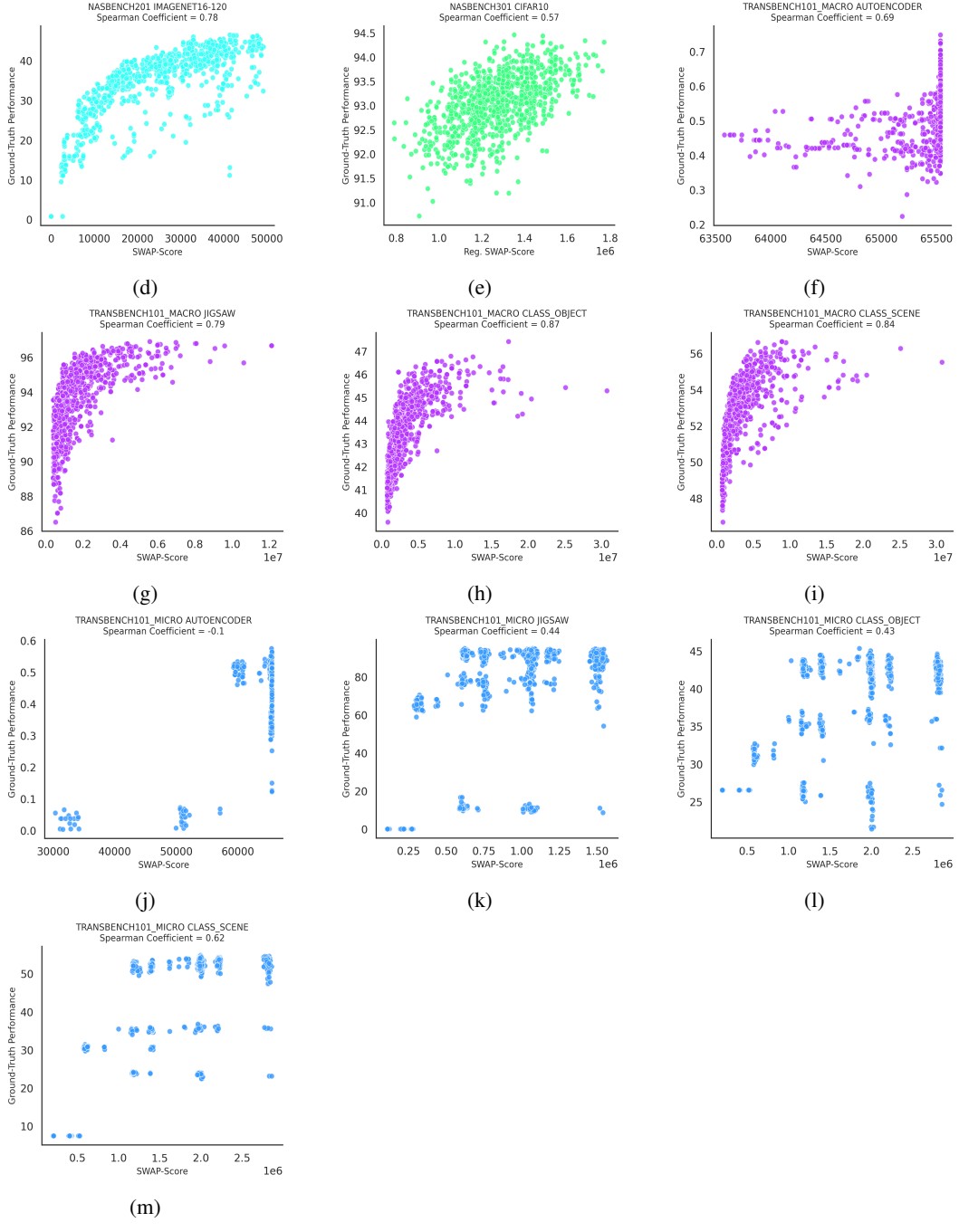

Figure 16: Visualisation of correlation between SWAP-Score and networks' ground-truth performance, for each search space and task combination. Colors indicate the search spaces.

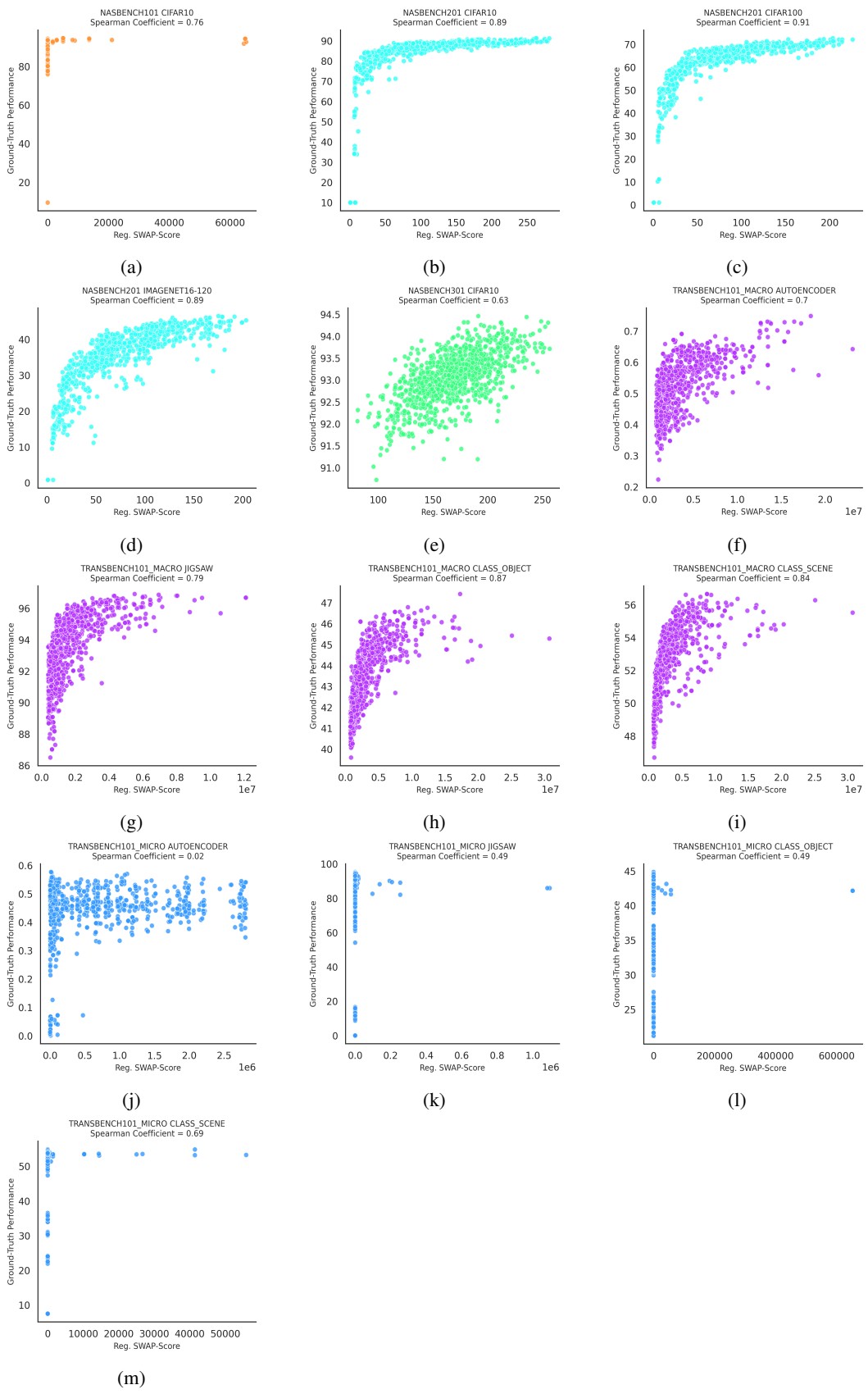

Figure 17: Visualisation of correlation between regularised SWAP-Score and networks' ground-truth performance, for each search space and task combination. Colors indicate the search spaces.

## F ARCHITECTURES FOUND BY SWAP-NAS ON DARTS SEARCH SPACE

Figures 18 to 21 demonstrate the neural architectures discovered by SWAP-NAS under varying model size constraints for the CIFAR-10 and ImageNet datasets. These figures effectively demonstrate the capability of the regularised SWAP-Score to control model size within the context of NAS. Additionally, they highlight the trend of increasing topological complexity in the architectures as the model size grows.

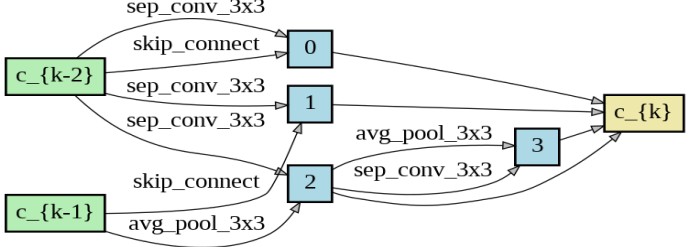

Figure 18: Cell architecture found by SWAP-NAS on CIFAR-10 dataset with model size 3.06 MB.

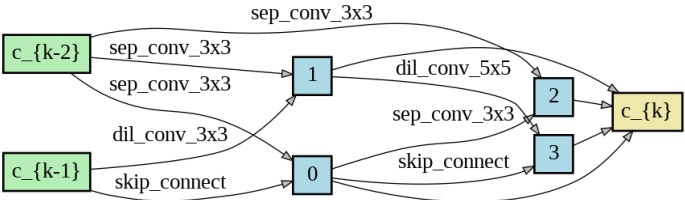

Figure 19: Cell architecture found by SWAP-NAS on CIFAR-10 dataset with model size 3.48 MB.

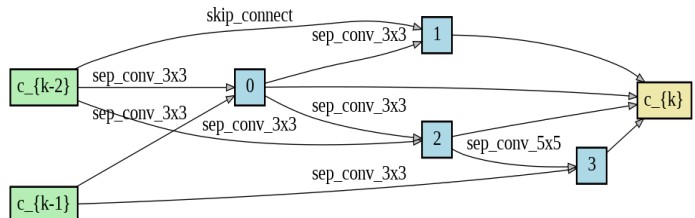

Figure 20: Cell architecture found by SWAP-NAS on CIFAR-10 dataset with model size 4.3 MB.

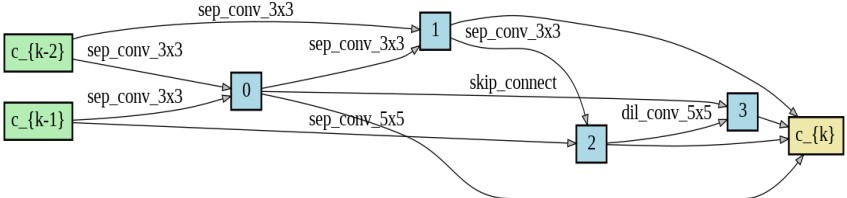

Figure 21: Cell architecture found by SWAP-NAS on ImageNet dataset with model size 5.8 MB.

