# OpenReview forum: "SWAP-NAS: Sample-Wise Activation Patterns for Ultra-fast NAS"
_ICLR.cc/2024/Conference — ICLR 2024 spotlight_

### Official Review · Reviewer_6s7y · 2023-10-13

**Soundness:** 3 good
**Presentation:** 3 good
**Contribution:** 3 good
**Rating:** 8
**Confidence:** 2

**Summary:**

In this paper, the authors introduced a novel training-free network evaluation metric based on Sample-Wise Activation Patterns. The proposed SWAP-Score and regularised SWAP-Score show stronger correlations with ground-truth performance than 15 existing training-free metrics on different spaces, stack-based and cell-based, for different tasks. The regularised SWAP-Score can enable model size
control during search and can further improve correlation in cell-based search spaces. It is integrated with an evolutionary search algorithm as SWAP-NAS.

**Strengths:**

The training-free NAS method shows better performance than previous SOTA and 5 NAS benchmarks. The searching time is reasonably low. The proposed metric looks very promising for model expressivity.

**Weaknesses:**

The method assumes ReLU as the activation function, which is a big limitation as more and more transformer based networks use GELU.

**Questions:**

Can this method apply to transformer-based NAS?

---

> ### Author Response · Authors · 2023-11-17
> **Response to Reviewer 6s7y**
>
> Thank you sincerely for the encouragement and the recognition in our work.  We appreciate your great effort and address your comments in detail below.
>
> **Weaknesses**
> > 1. The method assumes ReLU as the activation function, which is a big limitation as more and more transformer based networks use GELU.
>
> We totally agree that SWAP should be extended beyond ReLU, e.g. Transformers with GELU activation functions. This forms a key part of our ongoing work. As GELU is not a piecewise non-linear function like ReLU, the primary focus of our new work is on how to binarise the post-activation value from GELU. Unlike adopting the Signum as the indicator function for ReLU networks, we need to design a different suitable indicator function, which can binarise activation values to 0 and 1.
>
> For this study, the challenge is the absence of dedicated NAS benchmarks for Transformers, akin to NAS-Bench-X01 for ReLU networks. To rigorously verify the applicability and effectiveness of SWAP-Score on GELU activation, as we did with ReLU networks in this work, we will prepare and train a large variety of Transformers to obtain their ground-truth performance. This process is undoubtedly resource intensive. We are actively working in this direction and hope to be able to present it as a complete but separate study real soon.
>
> **Questions**
> > 1. Can this method apply to transformer-based NAS?
>
> In a nutshell, it is highly likely. We are working on this. Please see more detailed explanation above.

---

### Official Review · Reviewer_QHog · 2023-10-30

**Soundness:** 4 excellent
**Presentation:** 4 excellent
**Contribution:** 3 good
**Rating:** 8
**Confidence:** 5

**Summary:**

This paper proposes a novel training-free metric called SWAP-Score, based on Sample-Wise Activation Patterns, for zero-shot Neural Architecture Search (NAS). The SWAP-Score measures the expressivity of networks over a batch of input samples and outperforms 15 existing training-free metrics on various search spaces and tasks. The authors also introduce regularisation to the SWAP-Score for model size control during the search

**Strengths:**

- Robust Experimental Results: Upon meticulously examining and executing the code provided in the supplementary files, I affirmed the robustness of the experimental results by myself. The effectiveness of the proposed SWAP-NAS has been convincingly validated through my independent verification.

- Comprehensive Experimental Validation: The authors have conducted extensive experiments across multiple NAS-Bench datasets, demonstrating the clear superiority of SWAP-NAS. This broad-spectrum analysis significantly strengthens the validity and generalizability of the proposed approach.

- Clarity and Accessibility of Presentation: The paper is exceptionally well-crafted, featuring lucid and coherent exposition. The methodology is articulated in a manner that is not only easy to comprehend but also straightforward to follow. This enhances the paper's accessibility to a broader audience within the field.

**Weaknesses:**

- Limited Applicability to ReLU-based Networks: One limitation of this approach is its dependency on neural networks using ReLU activations. While ReLU is widely used, it represents a subset of possible activation functions, which narrows the scope of application and might not encompass the full diversity of network architectures.

- Similarity to NWOT Format: It's worth noting that the format of the proposed zero-cost proxy shares similarities with existing methods like NWOT. While not necessarily a weakness, this overlap should be acknowledged and explored further to understand the distinctions and innovations brought by SWAP-NAS.

- Unclear Impact of Dimension: The influence of dimensionality remains somewhat obscure in the paper. It's important to consider that neural networks, especially in shallow layers, may exhibit varying feature map sizes. A more detailed analysis of how these differences affect SWAP-NAS would enhance the completeness of the study.

- Absence of Correlation Visualization: The paper lacks visualizations that illustrate the correlation between SWAP and performance, which could provide valuable insights into the method's behavior and its implications for optimization. The inclusion of such visual aids would strengthen the paper's argument and help readers grasp the methodology's practical significance.

**Questions:**

1. How are `sigma` and `mu` related to model parameters, and what mechanisms allow their automatic configuration in SWAP-NAS?

2. In Figure 4, what causes TranBench101 to exhibit inferior performance compared to NAS-Bench-x01? Are there specific factors that contribute to this observed difference?

3. The paper claims versatility of SWAP-NAS across various tasks, including object detection, autoencoding, and the jigsaw puzzle. However, the primary focus seems to be on image classification results, such as ImageNet and CIFAR. Can the authors shed light on the reason for this specific focus? Furthermore, are there any plans or intentions to investigate SWAP's performance in the other mentioned domains in future work?

---

> ### Author Response · Authors · 2023-11-17
> **Response to Reviewer QHog (Part 1)**
>
> We are immensely grateful for your effort in reviewing our work, as well as for your encouragement, constructive suggestions and insightful questions. In the following, we will address each of your suggestions and concerns point-by-point.
>
> **Weaknesses**
>
> > 1. Limited Applicability to ReLU-based Networks: One limitation of this approach is its dependency on neural networks using ReLU activations. While ReLU is widely used, it represents a subset of possible activation functions, which narrows the scope of application and might not encompass the full diversity of network architectures.
>
> We completely acknowledge this point as extending beyond ReLU has been in our plan for quite a while. This work specifically focuses on ReLU-based networks due to several reasons. Firstly, the proposed SWAP concept is derived from studies on ReLU-based networks, especially in terms of network performance estimation. Secondly, we aim to address the unsolved limitations of these studies, such as the saturation problem (reached the upper bound) and poor correlation. Additionally, as you mentioned, NAS benchmarks and algorithms are predominately ReLU based, making it more feasible for large-scale comparisons, especially compared with other training-free metrics.
>
> We fully recognise the significance of expanding SWAP, e.g. to Transformers with GELU. This is indeed our ongoing work. Applying SWAP on a certain kind of activation function primarily depends on the indicator function, such as adopting Signum to binarise the activation values from ReLU. The major challenge of this extension is the lack of established NAS benchmarks for non-ReLU networks, and the poor availability of trained networks. Nevertheless, we are committed to exploring the effectiveness and generalization of SWAP across different activation functions and networks, aiming to present it as a complete but separate study.
>
> > 2. Similarity to NWOT Format: It's worth noting that the format of the proposed zero-cost proxy shares similarities with existing methods like NWOT. While not necessarily a weakness, this overlap should be acknowledged and explored further to understand the distinctions and innovations brought by SWAP-NAS.
>
> Thank you for pointing that out. NWOT and SWAP indeed share some similarities, as both methods aim to estimate networks’ performance by examining activation patterns. However, there are fundamental differences in their methodologies. In NWOT, authors first calculate the Hamming distance between pair of binary codes (rows in the standard activation patterns), they called it the activation overlap between data points (as we mentioned in the Related Work section in Page 3). Then, they use the Hamming distances to form a kernel matrix $K_{H}$. Finally, the score $s$ is obtained by computing the logarithm of the determinant of the kernel matrix $K_{H}$.
>
> In contrast, SWAP-Score is obtained by computing the cardinality of sample-wise activation patterns. This process is less computationally intensive than the NWOT score calculation. In addition, the proposed sample-wise activation patterns dose not only address the limitation of easitimating networks’ performance by standard activation patterns, but also introduces a novel paradigm for subsequent work. Specifically, SWAP is a new form of activation patterns that offer higher capacity for distinct patterns while preserving the information from standard activation patterns. We see it as a distinctive innovation of significant value to the community.
>
> If the above explanation adds value, we will include a brief version in the revised manuscript.
>
> > 3. Unclear Impact of Dimension: The influence of dimensionality remains somewhat obscure in the paper. It's important to consider that neural networks, especially in shallow layers, may exhibit varying feature map sizes. A more detailed analysis of how these differences affect SWAP-NAS would enhance the completeness of the study.
>
> Thank you for pointing this out. We did study the impact of dimensionality of the inputs. The results are included in Appendix D (Fig. 15). As mentioned in Section 3.2, the feature map size is impacted by either dimensionality of input samples or the depth of the neural networks. Although shallow layers may yield varying feature map sizes, the total size of the feature map from a given network is constant, unless either the dimensionality of the input samples or the architecture of network changes. Further, as per the results shown in Fig. 15, both SWAP-Score and regularized SWAP-Score show good and robust correlation with the ground-truth performance, no matter the varying dimensionality of inputs. In comparison, the standard activation pattern suffers greatly with the increase of dimensionality (the green line).

---

> ### Author Response · Authors · 2023-11-17
> **Response to Reviewer QHog (Part 2)**
>
> > 4. Absence of Correlation Visualization: The paper lacks visualizations that illustrate the correlation between SWAP and performance, which could provide valuable insights into the method's behavior and its implications for optimization. The inclusion of such visual aids would strengthen the paper's argument and help readers grasp the methodology's practical significance.
>
> Thank you for the nice suggestion, we have included the correlation visualisations as Fig. 16 & 17 in Appendix E of the revised version.
>
> **Questions**
> > 1. How are sigma and mu related to model parameters, and what mechanisms allow their automatic configuration in SWAP-NAS?
>
> The impact of $\mu$ and $\sigma$ are explained in Fig. 6, Appendix A. $\mu$ controls the center of the bell curve, thus, a model with size closer to $\mu$ will have less reduction in its SWAP-Score. $\sigma$ controls the shape of the bell curve, adjusting the tolerance of the regularisation.
>
> Both $\sigma$ and $\mu$ are automatically set by examining the distribution of model sizes of 1000 randomly sampled architectures. This process happens before the architecture search in an end-to-end manner, and only takes a few seconds. They can be manually pre-defined as well, if the goal is to find networks with a certain model size.
>
> > 2. In Figure 4, what causes TranBench101 to exhibit inferior performance compared to NAS-Bench-x01? Are there specific factors that contribute to this observed difference?
>
> Excellent observation. The problem here mainly occurs in the TransBench101-Micro space. TransBench101 includes two types of search spaces: micro-level and macro-level. The micro-level space consists of shallow networks (cells), the final models are constructed by repeated building blocks (cells). The macro-level space consists of the complete networks, which are the final models themselves. Thus, SWAP-Scores show good correlations on TransBench101-Macro space. In contrast, on the TransBench101-Micro space, the SWAP-Scores are calculated on cell networks not the final models. Hence, the correlation of SWAP-Scores is calculated based on a cell’s SWAP-Scores and the final model’s performance. The assumption is the performance of the final models is not simply equivalent to the performance of the cell network, especially on tasks beyond image classification. However, TransBench101 only provided the ground-truth performance of the final models rather than cell networks, so this assumption cannot be verified yet. We can investigate this phenomenon and delve into it deeper in future work.
>
> > 3. The paper claims versatility of SWAP-NAS across various tasks, including object detection, autoencoding, and the jigsaw puzzle. However, the primary focus seems to be on image classification results, such as ImageNet and CIFAR. Can the authors shed light on the reason for this specific focus? Furthermore, are there any plans or intentions to investigate SWAP's performance in the other mentioned domains in future work?
>
> For tasks beyond image classification, we mainly indicate the experiments on the benchmark spaces. As for the reason behind CIFAR and ImageNet, it is primarily for comprehensive comparison as they are the widely adopted tasks in NAS literature. The goal of this study is to propose SWAP and demonstrate its effectiveness by highlighting its performance on common NAS tasks. The adoption of CIFAR and ImageNet enables us to evaluate SWAP-NAS against the majority of SoTA NAS methods.
>
> Indeed, we have already started expanding SWAP beyond this study, e.g., different spaces, activation functions and tasks.

---

> > ### Comment · Reviewer_QHog · 2023-11-20
> > **Elevated Rating for Excellent Rebuttal**
> >
> > Excellent work! Swap's simplicity is indeed one of its strengths, and it has proven to be highly efficient. I am pleased to see that the authors have adequately addressed most of the concerns raised during the review process. Consequently, I have decided to upgrade my rating of this paper to 8 (accept).
> >
> > The limitation in ReLU-based Network and TransBench101 are actually hard for most of training-free methods. So I am looking forward to your further advancement in solving these problems.

---

> > > ### Author Response · Authors · 2023-11-21
> > > **Response to Reviewer QHog**
> > >
> > > We are very grateful for your encouragement and the rating increase. Looking forward to crossing paths very soon, especially on our new work.

---

### Official Review · Reviewer_AWYQ · 2023-10-31

**Soundness:** 3 good
**Presentation:** 3 good
**Contribution:** 3 good
**Rating:** 8
**Confidence:** 4

**Summary:**

This paper proposes a training-free metric called SWAP-score, which measures the expressivity of the networks. This metric can be integrated with the evolutionary NAS method to obtain competitive performance on several datasets with the least search cost.

**Strengths:**

1. The proposed metric is efficient and has a good correlation with the ground truth performance across various search spaces and tasks, beating all existing training-free metrics in public benchmarks.
2. The metric combined with the evolutionary search method can achieve good performance with extremely small search costs.

**Weaknesses:**

1. Missing references:
PINAT: A Permutation INvariance Augmented Transformer for NAS Predictor  AAAI 2023
TNASP: A Transformer-based NAS Predictor with a Self-evolution Framework  NeurIPS 2021
Please cite and compare with them in Table 1 and Table 2.
2. The method is proposed to be used in the ReLU activation function-based network, which is not been proven to work well on other nonlinear activation function-based networks or not.

**Questions:**

NA

---

> ### Author Response · Authors · 2023-11-17
> **Response to Reviewer AWYQ**
>
> Thank you greatly for dedicating your time and expertise to reviewing our paper. Your suggestions have not only been invaluable but have also broadened our horizons regarding the related literature. Please see our response below.
>
> **Weaknesses**
> > 1. Missing references: PINAT: A Permutation INvariance Augmented Transformer for NAS Predictor AAAI 2023 TNASP: A Transformer-based NAS Predictor with a Self-evolution Framework NeurIPS 2021 Please cite and compare with them in Table 1 and Table 2.
>
> Thanks for pointing that out, we have included them in Tables 1 & 2 of the revised version.
>
> > 2. The method is proposed to be used in the ReLU activation function-based network, which is not been proven to work well on other nonlinear activation function-based networks or not.
>
> Thank you for raising this issue. Going beyond ReLU has been in our plan since the early stage of this study and is now working in progress. As discussed in the future work section, verifying the effectiveness of SWAP-Score on other non-linear activation function-based networks is indeed crucial and valuable. We are actively pursuing this direction. The key point of extending SWAP to other kinds of activation function is adopting or developing a suitable indicator function, such as adopting the Signum function as an indicator function for the activation values from the ReLU in this work. The reason that we primarily focus on ReLU networks in this work is, the majority of popular NAS-Benchmarks, including the recently proposed NAS-Bench-Suite-Zero for validating training-free metrics, are predominantly built upon ReLU networks. These benchmarks form the foundation of our current study and provide solid platforms for large-scale validation experiments.
>
> Expanding our research to include networks with different activation functions, such as Transformers with GELU activations, involves a sizeable amount of work of its own. This entails more than developing a suitable indicator function, but preparation of a diverse range of Transformer architectures, obtaining their ground-truth performance on across a range of tasks. This is our ongoing work, and we would love to present it as a separate study in due time.

---

### Official Review · Reviewer_pab6 · 2023-11-06

**Soundness:** 3 good
**Presentation:** 3 good
**Contribution:** 2 fair
**Rating:** 6
**Confidence:** 3

**Summary:**

This paper proposes a new approach called SWAP-NAS for ultra-fast Neural Architecture Search (NAS) based on Sample-Wise Activation Patterns. SWAP-NAS outperforms existing metrics on various search spaces and tasks, achieving competitive performance on CIFAR-10 and ImageNet in just a few minutes of GPU time. The paper also discusses the limitations of existing training-free metrics in NAS and how regularisation can enhance the SWAP-Score and enable model size control during the search.

**Strengths:**

Strengths:
- The proposed SWAP-Score and regularised SWAP-Score show much stronger correlations with ground-truth performance than existing training-free metrics on different spaces and tasks.
- The regularised SWAP-Score can enable model size control during search and can further improve correlation in cell-based search spaces.
- When integrated with an evolutionary search algorithm as SWAP-NAS, a combination of ultra-fast architecture search and highly competitive performance can be achieved on both CIFAR-10 and ImageNet, outperforming SoTA NAS methods.

**Weaknesses:**

- The paper only give numerical numbers on NAS Search Space, and did not show or analyze any searched neural architectures, to back the claim that Sample-Wise Activation Patterns would measure the network’s expressivity more accurately.

**Questions:**

- How would SWAP-NAS integrate with pruning-based search method such as the one used in TE-NAS?
- How to intergate regularised SWAP-NAS with FLOPs or Latency budget constrain?

---

> ### Author Response · Authors · 2023-11-17
> **Response to Reviewer pab6**
>
> We sincerely thank you for the time and effort you have invested in reviewing our paper. Your comments and questions are very insightful and highly valuable for us to enhance the paper.  Your questions are addressed point-by-point in the following.
>
> **Weaknesses**
> >The paper only give numerical numbers on NAS Search Space, and did not show or analyze any searched neural architectures, to back the claim that Sample-Wise Activation Patterns would measure the network’s expressivity more accurately.
>
> Thank you for the excellent suggestion. Accordingly, we have added detailed visualisations of the correlation between networks’ ground-truth performance and SWAP-Scores, for all the combinations of search spaces and tasks (Appendix E of the revised version). Further, we have added visualisations of neural architectures discovered by SWAP-NAS (Appendix F of the revised version). Those new visualisations can demonstrate the advantages of the proposed method more intuitively as per your suggestion.
>
> We would like to stress that the major focus of this work is to propose and validate the effectiveness of the SWAP-Score as a training-free metric. The networks’ expressivity is an abstract concept. Its specific manifestation is the ground-truth performance. As mentioned in the paper, a network with higher expressivity means it is more capable of learning complex functions. Consequently, this kind of network is more likely to perform better. Thus, the high correlation between SWAP-Scores and networks’ ground-truth performance is strong evidence that SWAP-Score can more accurately measure the networks’ expressivity (Fig. 4). Moreover, our experiments of validating the effectiveness of SWAP-Score were conducted through NAS-Bench-Suite-Zero, which is a standardized framework designed to facilitate a fair comparison for different training-free metrics.
>
> **Questions**
>
> >1. How would SWAP-NAS integrate with pruning-based search method such as the one used in TE-NAS?
>
> The integration of SWAP-Score with pruning and other non-evolutionary search methods is indeed feasible and potentially valuable.  As both SWAP-NAS and TE-NAS adopt DARTS space, there is no difference from the search space perspective, except one is via an evolution-based search algorithm and the other is via a pruning-based. Notably, one of the training-free metrics utilised in TE-NAS, the number of linear regions, serves as a training-free metric via revealing the network’s expressivity as well. Thus, the transition from using the number of linear regions to SWAP-Score in a pruning-based NAS is straightforward. We are grateful for the suggestion and are indeed open to exploring the application of SWAP-Score in a wider range of search algorithms in our future work.
>
> >2. How to integrate regularised SWAP-NAS with FLOPs or Latency budget constrain?
>
> We see no obvious obstacle in the integration of SWAP-Score with FLOPs or latency. Both model parameters and FLOPs can be obtained once the model is constructed. Thus, FLOPs can be incorporated into our regularisation function as the model parameters. Regarding latency, it can be efficiently measured by passing a sample input though the untrained network and measuring the time required for the forward pass. This measurement can be done in parallel with the calculation of SWAP-Score. The result can also be added into the regularization function. An alternative solution is using multi-objective search considering both SWAP-Score and latency at the same time.
>
> It is important to note that our use of a regularisation function to control model size stems from a widely recognised limitation of training-free metrics, which often favor larger models. This limitation has been underscored in several studies, such as [1, 2].
>
> Again, thank you for this insightful comment which is very valuable for our next phase of the study.
>
> &nbsp;
> &nbsp;
>
> ##### [1] Xuefei Ning, Changcheng Tang, Wenshuo Li, Zixuan Zhou, Shuang Liang, Huazhong Yang, and Yu Wang. Evaluating efficient performance estimators of neural architectures. Advances in Neural Information Processing Systems, 34:12265–12277, 2021.
>
> ##### [2] Arjun Krishnakumar, Colin White, Arber Zela, Renbo Tu, Mahmoud Safari, and Frank Hutter. Nas- bench-suite-zero: Accelerating research on zero cost proxies. In Thirty-sixth Conference on Neural Information Processing Systems Datasets and Benchmarks Track, 2022.

---

### Author Response · Authors · 2023-11-17
**General Response**

We wholeheartedly thank all of you for the great effort and the meticulous review, especially to the extent of examining and running our code. The comments and suggestions are truly impressive, very helpful to enhance our paper. In response to the feedback, we have made the following updates to the revised manuscript:

1. Two more recent publications, TNASP and PINAT, have now been incorporated into **Tables 1** and **2**. This further enhances our claim and provides a more comprehensive comparison to the recent literature.

2. Visualisations have been added to further illustrate the effectiveness of Sample-Wise Activation Patterns (SWAP), as **Figures 16** and **17** in **Appendix E**. These figures visualize the correlation between the ground-truth performance of candidate networks and their SWAP-Score and Regularized SWAP-Score, across different search spaces and different tasks. These visualisations demonstrate the significance of our method in a more intuitive way.

3. More figures have been added in **Appendix F**, to visualise neural architectures that discovered by SWAP-NAS, from the DARTS space for CIFAR-10 and ImageNet datasets. Such visualisations indeed enhance the claims of our study, as readers can easily see the topological differences between architectures of varying model sizes and the effectiveness of the proposed regularized SWAP score.

---

### Meta-Review · Area_Chair_Yubo · 2023-12-05

**Metareview:**

This paper proposes a new approach called SWAP-NAS for ultra-fast Neural Architecture Search (NAS) based on Sample-Wise Activation Patterns. SWAP-NAS outperforms existing metrics on various search spaces and tasks, achieving competitive performance on CIFAR-10 and ImageNet and the relevant benchmarks. The paper also discusses the limitations of existing training-free metrics in NAS and how regularisation can enhance the SWAP-Score and enable model size control during the search.

**Justification For Why Not Higher Score:**

The method is limited to networks with ReLU as non-linearity

**Justification For Why Not Lower Score:**

The paper received positive scores and the rebuttals are extensive and address the raised points.

---

### Decision · Program_Chairs · 2024-01-16

Accept (spotlight)